# Reintroducing Straight-Through Estimators as Principled Methods for Stochastic Binary Networks[*]

Alexander Shekhovtsov[1] and Viktor Yanush[2]

[1] Czech Technical University in Prague
`shekhole@fel.cvut.cz`
[2] Samsung-HSE Laboratory National Research University Higher School of Economics, Moscow
`yanushviktor@gmail.com`

**Abstract.** Training neural networks with binary weights and activations is a challenging problem due to the lack of gradients and difficulty of optimization over discrete weights. Many successful experimental results have been achieved with empirical straight-through (ST) approaches, proposing a variety of ad-hoc rules for propagating gradients through non-differentiable activations and updating discrete weights. At the same time, ST methods can be truly derived as estimators in the stochastic binary network (SBN) model with Bernoulli weights. We advance these derivations to a more complete and systematic study. We analyze properties, estimation accuracy, obtain different forms of correct ST estimators for activations and weights, explain existing empirical approaches and their shortcomings, explain how latent weights arise from the mirror descent method when optimizing over probabilities. This allows to reintroduce ST methods, long known empirically, as sound approximations, apply them with clarity and develop further improvements.

## 1  Introduction

Neural networks with binary weights and activations have much lower computation costs and memory consumption than their real-valued counterparts [18, 26, 45]. They are therefore very attractive for applications in mobile devices, robotics and other resource-limited settings, in particular for solving vision and speech recognition problems [8, 56].

The seminal works that showed feasibility of training networks with binary weights [15] and binary weights and activations [27] used the empirical straight-through gradient estimation approach. In this approach the derivative of a step function like sign, which is zero, is substituted with the derivative of some other function, hereafter called a *proxy* function, on the backward pass. One possible choice is to use *identity* proxy, *i.e.*, to completely bypass sign on the backward pass, hence the name *straight-through* [5]. This ad-hoc solution appears to work

---

[*] We gratefully acknowledge support by Czech OP VVV project "Research Center for Informatics (CZ.02.1.01/0.0/0.0/16019/0000765)"

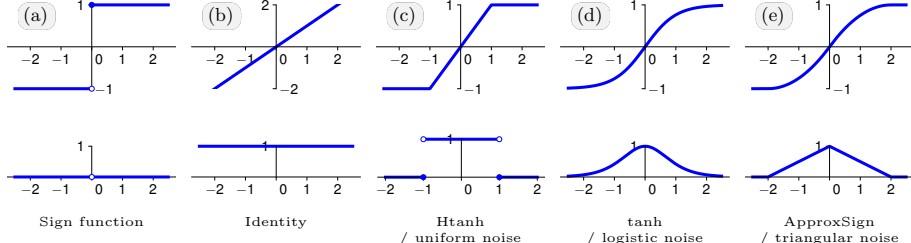

**Fig. 1:** The sign function and different proxy functions for derivatives used in empirical ST estimators. Variants (c-e) can be obtained by choosing the noise distribution in our framework. Specifically for a real-valued noise $z$ with cdf $F$, in the upper plots we show $\mathbb{E}_z[\text{sign}(a - z)] = 2F - 1$ and, respectively, twice the density, $2F'$ in the lower plots. Choosing *uniform* distribution for $z$ gives the density $p(z) = \frac{1}{2}\mathbb{1}_{z\in[-1,1]}$ and recovers the common Htanh proxy in (c). The *logistic* noise has cdf $F(z) = \sigma(2z)$, which recovers tanh proxy in (d). The *triangular* noise has density $p(z) = \max(0, |(2 - x)/4|)$, which recovers a scaled version of ApproxSign [34] in (e). The scaling (standard deviation) of the noise in each case is chosen so that $2F'(0) = 1$. The identity ST form in (b) we recover as latent weight updates with mirror descent.

surprisingly well and the later mainstream research on binary neural networks heavily relies on it [2, 6, 9, 11, 18, 34, 36, 45, 52, 60].

The *de-facto standard* straight-through approach in the above mentioned works is to use deterministic binarization and the clipped identity proxy as proposed by Hubara et al. [27]. However, other proxy functions were experimentally tried, including tanh and piece-wise quadratic ApproxSign [18, 34], illustrated in Fig. 1. This gives rise to a diversity of empirical ST methods, where various choices are studied purely experimentally [2, 6, 52]. Since binary weights can be also represented as a sign mapping of some real-valued *latent weights*, the same type of methods is applied to weights. However, often a different proxy is used for the weights, producing additional unclear choices. The dynamics and interpretation of latent weights are also studied purely empirically [51]. With such obscurity of latent weights, Helwegen et al. [24] argues that "latent weights do not exist" meaning that discrete optimization over binary weights needs to be considered. The existing partial justifications of deterministic straight-through approaches are limited to one-layer networks with Gaussian data [58] or binarization of weights only [1] and do not lead to practical recommendations.

In contrast to the deterministic variant used by the mainstream SOTA, straight-through methods were originally proposed (also empirically) for stochastic autoencoders [25] and studied in models with stochastic binary neurons [5, 44]. In the *stochastic binary network* (SBN) model which we consider, all hidden units and/or weights are Bernoulli random variables. The expected loss is a truly differentiable function of parameters (*i.e.*, weight probabilities) and its gradient can be estimated. This framework allows to pose questions such as: "What is the true expected gradient?" and "How far from it is the estimate computed by

ST?" Towards computing the true gradient, unbiased gradient estimators were developed [20, 55, 57], which however have not been applied to networks with deep binary dependencies due to increased variance in deep layers and complexity that grows quadratically with the number of layers [48]. Towards explaining ST methods in SBNs, Tokui & Sato [54] and Shekhovtsov et al. [48] showed how to *derive* ST under linearizing approximations in SBNs. These results however were secondary in these works, obtained from more complex methods. They remained unnoticed in the works applying ST in practice and recent works on its analysis [13, 58]. They are not properly related to existing empirical ST variants for activations and weights and did not propose analysis.

The goal of this work is to reintroduce straight-through estimators in a principled way in SBNs, to formalize and systematize empirical ST approaches for activation and weights in shallow and deep models. Towards this goal we review the derivation and formalize many empirical variants and algorithms using the derived method and sound optimization frameworks: we show how different kinds of ST estimators can occur as valid modeling choices or valid optimization choices. We further study properties of ST estimator and its utility for optimization: we theoretically predict and experimentally verify the improvement of accuracy with network width and show that popular modifications such as deterministic ST decrease this accuracy. For deep SBNs with binary weights we demonstrate that several estimators lead to equivalent results, as long as they are applied consistently with the model and the optimization algorithm.

More details on the related work, including alternative approaches for SBNs we discuss in Appendix A.

## 2   Derivation and Analysis

**Notation** We model random states $\boldsymbol{x} \in \{-1, 1\}^n$ using the noisy sign mapping:

$$x_i = \text{sign}(a_i - z_i), \tag{1}$$

where $z_i$ are real-valued independent noises with a fixed cdf $F$ and $a_i$ are (input-dependent) parameters. Equivalently to (1), we can say that $x_i$ follows $\{-1, 1\}$ valued Bernoulli distribution with probability $p(x_i{=}1) = \mathbb{P}(a_i{-}z_i \geq 0) = \mathbb{P}(z_i \leq a_i) = F(a_i)$. The noise cdf $F$ will play an important role in understanding different schemes. For logistic noise, its cdf $F$ is the logistic sigmoid function $\sigma$.

**Derivation** Straight-through method was first proposed empirically [25, 32] in the context of stochastic autoencoders, highly relevant to date [*e.g.* 16]. In contrast to more recent works applying variants of deterministic ST methods, these earlier works considered stochastic networks. It turns out that in this context it is possible to derive ST estimators exactly in the same form as originally proposed by Hinton. This is why we will first derive, using observations of [48, 54], analyze and verify it for stochastic autoencoders.

Let $\boldsymbol{y}$ denote observed variables. The *encoder network*, parametrized by $\boldsymbol{\phi}$, computes logits $\boldsymbol{a}(\boldsymbol{y}; \boldsymbol{\phi})$ and samples a binary latent state $\boldsymbol{x}$ via (1). As noises $\boldsymbol{z}$

| **Algorithm 1:** Straight-Through-Activations | **Algorithm 2:** Straight-Through-Weights |
|---|---|

```
/* a: preactivation        */      /* η: latent weights       */
/* F: injected noise cdf   */      /* F: weight noise cdf      */
/* x ∈ {-1,1}ⁿ             */      /* w ∈ {-1,1}ᵈ             */
1 Forward( a )                      1 Forward( η )
2 |   p = F(a);                     2 |   p = F(η);
3 |   return x ∼ 2Bernoulli(p) − 1; 3 |   return w ∼ 2Bernoulli(p) − 1;
4 Backward( dℒ/dx )                 4 Backward( dℒ/dw )
5 |   return dℒ/da ≡ 2 diag(F'(a)) dℒ/dx; 5 |   return dℒ/dη ≡ 2 dℒ/dw;
```

are independent, the conditional distribution of hidden states given observations $p(\boldsymbol{x}|\boldsymbol{y};\boldsymbol{\phi})$ factors as $\prod_i p(x_i|\boldsymbol{y};\boldsymbol{\phi})$. The *decoder* reconstructs observations with $p^{\mathrm{dec}}(\boldsymbol{y}|\boldsymbol{x};\boldsymbol{\theta})$ — another neural network parametrized by $\boldsymbol{\theta}$. The autoencoder reconstruction loss is defined as

$$\mathbb{E}_{\boldsymbol{y}\sim\mathrm{data}}\big[\mathbb{E}_{\boldsymbol{x}\sim p(\boldsymbol{x}|\boldsymbol{y};\boldsymbol{\phi})}[-\log p^{\mathrm{dec}}(\boldsymbol{y}|\boldsymbol{x};\boldsymbol{\theta})]\big]. \tag{2}$$

The main challenge is in estimating the gradient w.r.t. the encoder parameters $\boldsymbol{\phi}$ (differentiation in $\boldsymbol{\theta}$ can be simply taken under the expectation). The problem for a fixed observation $\boldsymbol{y}$ takes the form

$$\tfrac{\partial}{\partial\boldsymbol{\phi}}\mathbb{E}_{\boldsymbol{x}\sim p(\boldsymbol{x};\boldsymbol{\phi})}[\mathcal{L}(\boldsymbol{x})] = \tfrac{\partial}{\partial\boldsymbol{\phi}}\mathbb{E}_{\boldsymbol{z}}[\mathcal{L}(\mathrm{sign}(\boldsymbol{a}-\boldsymbol{z}))], \tag{3}$$

where $p(\boldsymbol{x};\boldsymbol{\phi})$ is a shorthand for $p(\boldsymbol{x}|\boldsymbol{y};\boldsymbol{\phi})$ and $\mathcal{L}(\boldsymbol{x}) = -\log p^{\mathrm{dec}}(\boldsymbol{y}|\boldsymbol{x};\boldsymbol{\theta})$. The reparametrization trick, *i.e.*, to draw one sample of $\boldsymbol{z}$ in (3) and differentiate $\mathcal{L}(\mathrm{sign}(\boldsymbol{a}-\boldsymbol{z}))$ fails: since the loss as a function of $\boldsymbol{a}$ and $\boldsymbol{z}$ is not *continuously differentiable we cannot interchange the gradient and the expectation in $\boldsymbol{z}$*[3]. If we nevertheless attempt the interchange, we obtain that the gradient of $\mathrm{sign}(\boldsymbol{a}-\boldsymbol{z})$ is zero as well as its expectation. Instead, the following steps lead to an unbiased low-variance estimator. From the LHS of (3) we express the derivative as

$$\tfrac{\partial}{\partial\boldsymbol{\phi}}\sum_{\boldsymbol{x}}\big(\prod_i p(x_i;\boldsymbol{\phi})\big)\mathcal{L}(\boldsymbol{x}) = \sum_{\boldsymbol{x}}\sum_i\big(\prod_{i'\neq i} p(x_{i'};\boldsymbol{\phi})\big)\big(\tfrac{\partial}{\partial\boldsymbol{\phi}}p(x_i;\boldsymbol{\phi})\big)\mathcal{L}(\boldsymbol{x}). \tag{4}$$

Then we apply *derandomization* [40, ch. 8.7], which performs summation over $x_i$ holding the rest of the state $\boldsymbol{x}$ fixed. Because $x_i$ takes only two values, we have

$$\sum_{x_i}\tfrac{\partial p(x_i;\boldsymbol{\phi})}{\partial\boldsymbol{\phi}}\mathcal{L}(\boldsymbol{x}) = \tfrac{\partial p(x_i;\boldsymbol{\phi})}{\partial\boldsymbol{\phi}}\mathcal{L}(\boldsymbol{x}) + \tfrac{\partial(1-p(x_i;\boldsymbol{\phi}))}{\partial\boldsymbol{\phi}}\mathcal{L}(\boldsymbol{x}_{\downarrow i})$$
$$= \tfrac{\partial}{\partial\boldsymbol{\phi}}p(x_i;\boldsymbol{\phi})\big(\mathcal{L}(\boldsymbol{x}) - \mathcal{L}(\boldsymbol{x}_{\downarrow i})\big), \tag{5}$$

where $\boldsymbol{x}_{\downarrow i}$ denotes the full state vector $\boldsymbol{x}$ with the sign of $x_i$ flipped. Since this expression is now invariant of $x_i$, we can multiply it with $1 = \sum_{x_i} p(x_i;\boldsymbol{\phi})$ and

---

[3] The conditions allow to apply Leibniz integral rule to exchange derivative and integral. Other conditions may suffice, *e.g.*, when using weak derivatives [17].

express the gradient (4) in the form:

$$\sum_i \sum_{\boldsymbol{x}_{\neg i}} \big( \textstyle\prod_{i' \neq i} p(x_{i'}; \boldsymbol{\phi}) \big) \sum_{x_i} p(x_i; \boldsymbol{\phi}) \frac{\partial p(x_i; \boldsymbol{\phi})}{\partial \boldsymbol{\phi}} \big( \mathcal{L}(\boldsymbol{x}) - \mathcal{L}(\boldsymbol{x}_{\downarrow i}) \big)$$

$$\sum_{\boldsymbol{x}} \big( \textstyle\prod_{i'} p(x_{i'}; \boldsymbol{\phi}) \big) \sum_i \frac{\partial p(x_i; \boldsymbol{\phi})}{\partial \boldsymbol{\phi}} \big( \mathcal{L}(\boldsymbol{x}) - \mathcal{L}(\boldsymbol{x}_{\downarrow i}) \big)$$

$$= \mathbb{E}_{\boldsymbol{x} \sim p(\boldsymbol{x}; \boldsymbol{\phi})} \sum_i \frac{\partial p(x_i, \boldsymbol{\phi})}{\partial \boldsymbol{\phi}} \big( \mathcal{L}(\boldsymbol{x}) - \mathcal{L}(\boldsymbol{x}_{\downarrow i}) \big), \qquad (6)$$

where $\boldsymbol{x}_{\neg i}$ denotes all states excluding $x_i$. To obtain an unbiased estimate, it suffices to take one sample $\boldsymbol{x} \sim p(\boldsymbol{x}; \boldsymbol{\phi})$ and compute the sum in $i$ in (6). This estimator is known as *local expectations* [53] and coincides in this case with GO-gradient [14], RAM [54] and PSA [48].

However, evaluating $\mathcal{L}(\boldsymbol{x}_{\downarrow i})$ for all $i$ may be impractical. A huge simplification is obtained if we assume that the change of the loss $\mathcal{L}$ when only a single latent bit $x_i$ is changed can be approximated via linearization. Assuming that $\mathcal{L}$ is defined as a differentiable mapping $\mathbb{R}^n \to \mathbb{R}$ (*i.e.*, that the loss is built up of arithmetic operations and differentiable functions), we can approximate

$$\mathcal{L}(\boldsymbol{x}) - \mathcal{L}(\boldsymbol{x}_{\downarrow i}) \approx 2x_i \frac{\partial \mathcal{L}(\boldsymbol{x})}{\partial x_i}, \qquad (7)$$

where we used the identity $x_i - (-x_i) = 2x_i$. Expanding the derivative of conditional density $\frac{\partial}{\partial \boldsymbol{\phi}} p(x_i; \boldsymbol{\phi}) = x_i F'(a_i(\boldsymbol{\phi})) \frac{\partial}{\partial \boldsymbol{\phi}} a_i(\boldsymbol{\phi})$, we obtain

$$\frac{\partial p(x_i, \boldsymbol{\phi})}{\partial \boldsymbol{\phi}} \big( \mathcal{L}(\boldsymbol{x}) - \mathcal{L}(\boldsymbol{x}_{\downarrow i}) \big) \approx 2F'(a_i(\boldsymbol{\phi})) \frac{\partial a_i(\boldsymbol{\phi})}{\partial \boldsymbol{\phi}} \frac{\partial \mathcal{L}(\boldsymbol{x})}{\partial x_i}. \qquad (8)$$

If we now define that $\frac{\partial x_i}{\partial a_i} \equiv 2F'(a_i)$, the summation over $i$ in (6) with the approximation (8) can be written in the form of a chain rule:

$$\sum_i 2F'(a_i(\boldsymbol{\phi})) \frac{\partial a_i(\boldsymbol{\phi})}{\partial \boldsymbol{\phi}} \frac{\partial \mathcal{L}(\boldsymbol{x})}{\partial x_i} = \sum_i \frac{\partial \mathcal{L}(\boldsymbol{x})}{\partial x_i} \frac{\partial x_i}{\partial a_i} \frac{\partial a_i(\boldsymbol{\phi})}{\partial \boldsymbol{\phi}}. \qquad (9)$$

To clarify, the estimator is already defined by the LHS of (9). We simply want to compute this expression by (ab)using the standard tools, and this is the sole purpose of introducing $\frac{\partial x_i}{\partial a_i}$. Indeed the RHS of (9) is a product of matrices that would occur in standard backpropagation. We thus obtained ST algorithm Alg. 1. We can observe that *it matches exactly to the one described by Hinton [25]: to sample on the forward pass and use the derivative of the noise cdf on the backward pass*, up to the multiplier 2 which occurred due to the use of $\pm 1$ encoding for $\boldsymbol{x}$.

## 2.1   Analysis

Next we study properties of the derived ST algorithm and its relation to empirical variants. We will denote a modification of Alg. 1 that does not use sampling in Line 3, but instead computes $x = \mathrm{sign}(a)$, a *deterministic ST*; and a modification that uses derivative of some other function $G$ instead of $F$ in Line 5 as *using a proxy $G$*.

**Invariances** Observe that binary activations (and hence the forward pass) stay invariant under transformations: $\text{sign}(a_i - z_i) = \text{sign}(T(a_i) - T(z_i))$ for any strictly monotone mapping $T$. Consistently, *the ST gradient by Alg. 1 is also invariant to $T$*. In contrast, empirical straight-through approaches, in which the derivative proxy is hand-designed, fail to maintain this property. In particular, rescaling the proxy leads to different estimators.

Furthermore, when applying transform $T = F$ (the noise cdf), the back-propagation rule in line 5 of Alg. 1 becomes equivalent to using the identity proxy. Hence we see that a common description of ST in the literature as "to back-propagate through the hard threshold function as if it had been the identity function" is also correct, *but only for the case of uniform noise* in $[-1, 1]$. Otherwise, and especially so for deterministic ST, this description is ambiguous because the resulting gradient estimator crucially depends on what transformations were applied under the hard threshold.

**ST Variants** Using the invariance property, many works applying randomized ST estimators are easily seen to be equivalent to Alg. 1: [16, 44, 49]. Furthermore, using different noise distributions for $\boldsymbol{z}$, we can obtain correct ST analogues for common choices of sign proxies used in empirical ST works as shown in Fig. 1 (c-e). In our framework they correspond to the choice of parametrization of the conditional Bernoulli distribution, which should be understood similarly to how a neural network can be parametrized in different ways.

If the "straight-through" idea is applied informally, however, this may lead to confusion and poor performance. The most cited reference for the ST estimator is Bengio et al. [5]. However, [5, Eq. 13] defines in fact the identity ST variant, incorrectly attributing it to Hinton (see Appendix A). We will show this variant to be less accurate for hidden units, both theoretically and experimentally. Pervez et al. [42] use $\pm 1$ binary encoding but apply ST estimator without coefficient 2. When such estimator is used in VAE, where the gradient of the prior KL divergence is computed analytically, it leads to a significant bias of the total gradient towards the prior. In Fig. 2 we illustrate that the difference in performance may be substantial. We analyze other techniques introduced in FouST in more detail in [47]. An inappropriate scaling by a factor of 2 can be as well detrimental in deep models, where the factor would be applied multiple times (in each layer).

**Bias Analysis** Given a rather crude linearization involved, it is indeed hard to obtain fine theoretical guarantees about the ST method. We propose an analysis targeting understanding the effect of common empirical variants and understanding conditions under which the estimator becomes more accurate. The respective formal theorems are given in Appendix B.

**I)** When ST is unbiased? As we used linearization as the only biased approximation, it follows that *Alg. 1 is unbiased if the objective function $\mathcal{L}$ is multilinear in $\boldsymbol{x}$*. A simple counter-example, where ST is biased, is $\mathcal{L}(x) = x^2$.

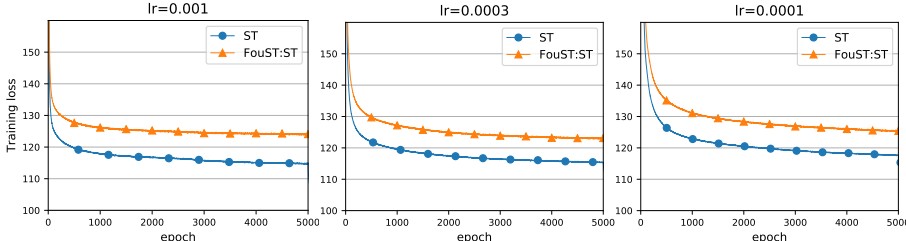

**Fig. 2:** Training VAE on MNIST, closely following experimental setup [42]. The plots show training loss (negative ELBO) during epochs for different learning rates. The variant of ST algorithm used [42] is misspecified because of the scaling factor and performs substantially worse at for all learning rates. Full experiment specification is given in Appendix D.1.

In this case the expected value of the loss is 1, independently of $a$ that determines $x$; and the true gradient is zero. However the expected ST gradient is $\mathbb{E}[2F'(a)2x] = 4F'(a)(2F(a) - 1)$, which may be positive or negative depending on $a$. On the other hand, any function of binary variables has an equivalent multilinear expression. In particular, if we consider $\mathcal{L}(\boldsymbol{x}) = \|\boldsymbol{W}\boldsymbol{x} - \boldsymbol{y}\|^2$, analyzed by Yin et al. [58], then $\tilde{\mathcal{L}}(\boldsymbol{x}) = \|\boldsymbol{W}\boldsymbol{x} - \boldsymbol{y}\|^2 - \sum_i x_i^2 \|\boldsymbol{W}_{:,i}\|^2 + \sum_i \|\boldsymbol{W}_{:,i}\|^2$ *coincides with $\mathcal{L}$ on all binary configurations and is multilinear.* It follows that ST applied to $\tilde{\mathcal{L}}$ gives an unbiased gradient estimate of $\mathbb{E}[\mathcal{L}]$, an immediate improvement compared to [58]. In the special case when $\mathcal{L}$ is linear in $\boldsymbol{x}$, the ST estimator is not only unbiased but has a zero variance, *i.e.*, it is exact.

**II)** How does using a mismatched proxy in Line 5 of Alg. 1 affect the gradient in $\boldsymbol{\phi}$? Since $\mathrm{diag}(F')$ occurs in the backward chain, we call estimators that use some matrix $\boldsymbol{\Lambda}$ instead of $\mathrm{diag}(F')$ as *internally rescaled*. We show that *for any $\boldsymbol{\Lambda} \succcurlyeq 0$, the expected rescaled estimator has non-negative scalar product with the expected original estimator.* Note that this is not completely obvious as the claim is about the final gradient in the model parameters $\boldsymbol{\phi}$ (*e.g.*, weights of the encoder network in the case of autoencoders). However, if the ST gradient by Alg. 1 is biased (when $\mathcal{L}$ is not multi-linear) but is nevertheless an ascent direction in expectation, the expected rescaled estimator may fail to be an ascent direction, *i.e.*, to have a positive scalar product with the true gradient.

**III)** When does ST gradient provide a valid ascent direction? Assuming that all partial derivatives $g_i(\boldsymbol{x}) = \frac{\partial \mathcal{L}(\boldsymbol{x})}{\partial x_i}$ are $L$-Lipschitz continuous for some $L$, we can show that *the expected ST gradient is an ascent direction for any network if and only if $\left| \mathbb{E}_{\boldsymbol{x}}[g_i(\boldsymbol{x})] \right| > L$ for all $i$.*

**IV)** Can we decrease the bias? Assume that the loss function is applied to a linear transform of Bernoulli variables, *i.e.*, takes the form $\mathcal{L}(\boldsymbol{x}) = \ell(\boldsymbol{W}\boldsymbol{x})$. A typical initialization uses random $\boldsymbol{W}$ normalized by the size of the fan-in, *i.e.*, such that $\|\boldsymbol{W}_{k,:}\|_2 = 1 \ \forall k$. In this case *the Lipschitz constant of gradients of $\mathcal{L}$ scales as $O(1/\sqrt{n})$, where $n$ is the number of binary variables.* Therefore, *using more binary variables decreases the bias, at least at initialization.*

**V)** Does deterministic ST give an ascent direction? Let $\boldsymbol{g}^*$ be the deterministic ST gradient for the state $\boldsymbol{x}^* = \text{sign}(\boldsymbol{a})$ and $p^* = p(\boldsymbol{x}^*|\boldsymbol{a})$ be its probability. *We show that deterministic ST gradient forms a positive scalar product with the expected ST gradient if $|g_i^*| \geq 2L(1-p^*)$ and with the true gradient if $|g_i^*| \geq 2L(1-p^*)+L$.* From this we conclude that deterministic ST positively correlates with the true gradient when $\mathcal{L}$ is multilinear, improves with the number of hidden units in the case described by IV and approaches expected stochastic ST as units learn to be deterministic so that the factor $(1-p^*)$ decreases.

**Deep ST** So far we derived and analyzed ST for a single layer model. It turns out that simply applying Alg. 1 in each layer of a deep model with conditional Bernoulli units gives the correct extension for this case. We will not focus on deriving deep ST here, but remark that it can be derived rigorously by chaining derandomization and linearization steps, discussed above, for each layer [48]. In particular, [48] show that ST can be obtained by making additional linearizations in their (more accurate) PSA method. The insights from the derivation are twofold. First, since derandomization is performed recurrently, the variance for deep layers is significantly reduced. Second, we know which approximations contribute to the bias, they are indeed the linearizations of all conditional Bernoulli probabilities in all layers and of the loss function as a function of the last Bernoulli layer. We may expect that using more units, similarly to property IV, would improve linearizing approximations of intermediate layers increasing the accuracy of deep ST gradient.

## 3   Latent Weights do Exist!

Responding to the work "Latent weights do not exist: Rethinking binarized neural network optimization" [24] and the lack of formal basis to introduce latent weights in the literature (*e.g.*, [27]), we show that such weights can be formally defined in SBNs and that several empirical update rules do in fact correspond to sound optimization schemes: projected gradient descent, mirror descent, variational Bayesian learning.

Let $\boldsymbol{w}$ be $\pm 1$-Bernoulli weights with $p(w_i{=}1) = \theta_i$, let $\mathcal{L}(\boldsymbol{w})$ be the loss function for a fixed training input. Consistently with the model for activations (1), we can define $w_i = \text{sign}(\eta_i - z_i)$ in order to model weights $w_i$ using parameters $\eta_i \in \mathbb{R}$ which we will call *latent weights*. It follows that $\theta_i = F_z(\eta_i)$. We need to tackle two problems in order to optimize $\mathbb{E}_{\boldsymbol{w}\sim p(\boldsymbol{w}|\boldsymbol{\theta})}[\mathcal{L}(\boldsymbol{w})]$ in probabilities $\boldsymbol{\theta}$: i) how to estimate the gradient and ii) how to handle constraints $\boldsymbol{\theta} \in [0,1]^m$.

**Projected Gradient** A basic approach to handle constraints is the *projected gradient descent*:

$$\boldsymbol{\theta}^{t+1} := \text{clip}(\boldsymbol{\theta}^t - \varepsilon \boldsymbol{g}^t, 0, 1), \tag{10}$$

where $\boldsymbol{g}^t = \nabla_{\boldsymbol{\theta}} \mathbb{E}_{\boldsymbol{w} \sim p(\boldsymbol{w}|\boldsymbol{\theta}^t)}[\mathcal{L}(\boldsymbol{w})]$ and $\mathrm{clip}(\boldsymbol{x}, a, b) := \max(\min(\boldsymbol{x}, b), a)$ is the projection. Observe that for the uniform noise distribution on $[-1, 1]$ with $F(z) = \mathrm{clip}(\frac{z+1}{2}, 0, 1)$, we have $\theta_i = p(w_i{=}1) = F(\eta_i) = \mathrm{clip}(\frac{\eta_i+1}{2}, 0, 1)$. Because this $F$ is linear on $[-1, 1]$, the update (10) can be equivalently reparametrized in $\boldsymbol{\eta}$ as

$$\boldsymbol{\eta}^{t+1} := \mathrm{clip}(\boldsymbol{\eta}^t - \varepsilon' \boldsymbol{h}^t, -1, 1), \tag{11}$$

where $\boldsymbol{h}^t = \nabla_{\boldsymbol{\eta}} \mathbb{E}_{\boldsymbol{w} \sim p(\boldsymbol{w}|F(\boldsymbol{\eta}))}[\mathcal{L}(\boldsymbol{w})]$ and $\varepsilon' = 4\varepsilon$. The gradient in the latent weights, $\boldsymbol{h}^t$, can be estimated by Alg. 1 and simplifies by expanding $2F' = 1$. We obtained that *the emperically proposed method of Hubara et al. [27, Alg.1] with stochastic rounding and with real-valued weights identified with $\boldsymbol{\eta}$ is equivalent to PGD on $\boldsymbol{\eta}$ with constraints $\eta \in [-1, 1]^m$ and ST gradient by Alg. 1.*

**Mirror Descent** As an alternative approach to handle constraints $\boldsymbol{\theta} \in [0, 1]^m$, we study the application of mirror descent (MD) and connect it with the identity ST update variants. A step of MD is found by solving the following proximal problem:

$$\boldsymbol{\theta}^{t+1} = \min_{\boldsymbol{\theta}} \left[ \langle \boldsymbol{g}^t, \boldsymbol{\theta} - \boldsymbol{\theta}^t \rangle + \tfrac{1}{\varepsilon} D(\boldsymbol{\theta}, \boldsymbol{\theta}^t) \right]. \tag{12}$$

The divergence term $\frac{1}{\varepsilon} D(\boldsymbol{\theta}, \boldsymbol{\theta}^t)$ weights how much we trust the linear approximation $\langle \boldsymbol{g}^t, \boldsymbol{\theta} - \boldsymbol{\theta}^t \rangle$ when considering a step from $\boldsymbol{\theta}^t$ to $\boldsymbol{\theta}$. When the gradient is stochastic we speak of *stochastic mirror descent* (SMD) [3, 59]. A common choice of divergence to handle probability constraints is the KL-divergence $D(\theta_i, \theta_i^t) = \mathrm{KL}(\mathrm{Ber}(\theta_i), \mathrm{Ber}(\theta_i^t)) = \theta_i \log(\frac{\theta_i}{\theta_i^t}) + (1 - \theta_i) \log(\frac{1-\theta_i}{1-\theta_i^t})$. Solving for a stationary point of (12) gives

$$0 = g_i^t + \tfrac{1}{\varepsilon}\left( \log(\tfrac{\theta_i}{1-\theta_i}) - \log(\tfrac{\theta_i^t}{1-\theta_i^t}) \right). \tag{13}$$

Observe that when $F = \sigma$ we have $\log(\frac{\theta_i}{1-\theta_i}) = \eta_i$. Then the MD step can be written in the well-known convenient form using the latent weights $\boldsymbol{\eta}$ (natural parameters of Bernoulli distribution):

$$\boldsymbol{\theta}^t := \sigma(\boldsymbol{\eta}^t); \qquad \boldsymbol{\eta}^{t+1} := \boldsymbol{\eta}^t - \varepsilon \nabla_{\boldsymbol{\theta}} \mathcal{L}(\boldsymbol{\theta}^t). \tag{14}$$

We thus have obtained the rule where on the forward pass $\boldsymbol{\theta} = \sigma(\boldsymbol{\eta})$ defines the sampling probability of $\boldsymbol{w}$ and on the backward pass the derivative of $\sigma$, that otherwise occurs in Line 5 of Alg. 1, *is bypassed exactly as if the identity proxy was used.* We define such ST rule for optimization in weights as Alg. 2. Its correctness is not limited to logistic noise. We show that for any strictly monotone noise distribution $F$ there is a corresponding divergence function $D$:

**Proposition 1.** *Common SGD in latent weights $\boldsymbol{\eta}$ using the* identity *straight-through-weights Alg. 2 implements SMD in the weight probabilities $\boldsymbol{\theta}$ with the divergence corresponding to $F$.*

Proof in Appendix C. Proposition 1 reveals that although Bernoulli weights can be modeled the same way as activations using the injected noise model

$w = \text{sign}(\eta - z)$, *the noise distribution F for weights correspond to the choice of the optimization proximity scheme.*

Despite generality of Proposition 1, we view the KL divergence as a more reliable choice in practice. Azizan et al. [3] have shown that the optimization with SMD has an inductive bias to find the closest solution to the initialization point as measured by the divergence used in MD, which has a strong impact on generalization. This suggests that MD with KL divergence will prefer higher entropy solutions, making more diverse predictions. It follows that SGD on latent weights with logistic noise and identity straight-through Alg. 2 enjoys the same properties.

**Variational Bayesian Learning** Extending the results above, we study the variational Bayesian learning formulation and show the following:

**Proposition 2.** *Common SGD in latent weights $\eta$ with a weight decay and identity straight-through-weights Alg. 2 is equivalent to optimizing a factorized variational approximation to the weight posterior $p(w|data)$ using a composite SMD method.*

Proof in Appendix C.2. As we can see, powerful and sound learning techniques can be obtained in a form of simple update rules using identity straight-through estimators. Therefore, identity-ST is fully rehabilitated in this context.

## 4    Experiments

**Stochastic Autoencoders** Previous work has demonstrated that Gumbel-Softmax (biased) and ARM (unbiased) estimators give better results than ST on training variational autoencoders with Bernoulli latents [16, 29, 57]. However, only the test performance was revealed to readers. We investigate in more detail what happens during training. Except of studying the training loss under the same training setup, we measure the gradient approximation accuracy using ARM with 1000 samples as the reference.

We train a simple yet realistic variant of stochastic autoencoder for the task of text retrieval with binary representation on *20newsgroups* dataset. The autoencoder is trained by minimizing the reconstruction loss (2). Please refer to Appendix D.2 for full specification of the model and experimental setup.

For each estimator we perform the following protocol. First, we train the model with this estimator using Adam with $lr = 0.001$ for 1000 epochs. We then switch the estimator to ARM with 10 samples and continue training for 500 more epochs (denoted as ARM-10 correction phase). Fig. 3 top shows the training performance for different number of latent bits $n$. It is seen (esp. for 8 and 64 bits) that some estimators (esp. ST and det_ST) appear to make no visible progress, and even increase the loss, while switching them to ARM makes a rapid improvement. Does it mean that these estimators are bad and ARM is very good? An explanation of this phenomenon is offered in Fig. 5. The rapid improvement by ARM is possible because these estimators have accumulated a

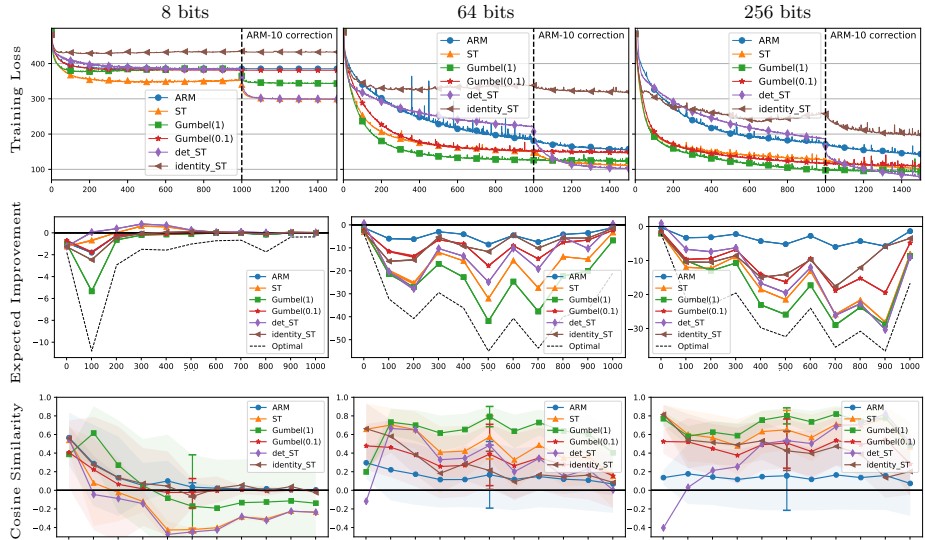

**Fig. 3:** Comparison of the training performance and gradient estimation accuracy for a stochastic autoencoder with different number of latent Bernoulli units (bits). *Training Loss:* each estimator is applied for 1000 epochs and then switched to ARM-10 in order to correct the accumulated bias. *Expected improvement*: lower is better (measures expected change of the loss), the dashed line shows the maximal possible improvement knowing the true gradient. *Cosine similarity*: higher is better, close to 1 means that the direction is accurate while below 0 means the estimated gradient is not an ascent direction; error bars indicate empirical 70% confidence intervals using 100 trials.

significant bias due to a systematic error component, which nevertheless can be easily corrected by an unbiased estimator.

To measure the bias and alignment of directions, as theoretically analyzed in Section 2.1, we evaluate different estimators at the same parameter points located along the learning trajectory of the reference ARM estimator. At each such point we estimate the true gradient $\boldsymbol{g}$ by ARM-1000. To measure the quality of a candidate 1-sample estimator $\tilde{\boldsymbol{g}}$ we compute the *expected cosine similarity* and the *expected improvement*, defined respectively as:

$$\text{ECS} = \mathbb{E}\Big[\langle \boldsymbol{g}, \tilde{\boldsymbol{g}}\rangle/(\|\boldsymbol{g}\|\|\tilde{\boldsymbol{g}}\|)\Big], \qquad \text{EI} = -\mathbb{E}[\langle \boldsymbol{g}, \tilde{\boldsymbol{g}}\rangle]/\sqrt{\mathbb{E}[\|\tilde{\boldsymbol{g}}\|^2]}, \tag{15}$$

The expectations are taken over 100 trials and all batches. A detailed explanation of these metrics is given in Appendix D.2. These measurements, displayed in Fig. 3 for different bit length, clearly show that with a small bit length biased estimators consistently run into producing wrong directions. *Identity ST and deterministic ST clearly introduce an extra bias to ST.* However, when we increase the number of latent bits, the accuracy of all biased estimators improves, confirming our analysis **IV**, **V**.

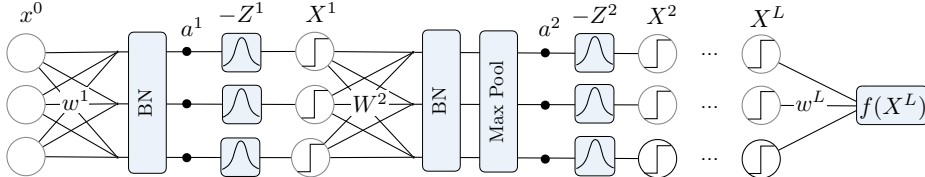

**Fig. 4:** Stochastic Binary Network: first and last layer have real-valued weights. BN layers have real-valued scale and bias parameters that can adjust scaling of activations relative to noise. $Z$ are independent injected noises with a chosen distribution. Binary weights $W_{ij}$ are random $\pm 1$ Bernoulli($\theta_{ij}$) with learnable probabilities $\theta_{ij}$. In experiments we consider SBN with a convolutional architecture same as [15, 27]: $(2 \times 128 \text{C}3) - \text{MP}2 - (2 \times 256 \text{C}3) - \text{MP}2 - (2 \times 512 \text{C}3) - \text{MP}2 - (2 \times 1024 \text{FC}) - 10 \text{FC} - \text{softmax}$.

The practical takeaways are as follows: 1) biased estimators may perform significantly better than unbiased but might require a correction of the systematically accumulated bias; 2) with more units the ST approximation clearly improves and the bias has a less detrimental effect, requiring less correction; 3) Alg. 1 is more accurate than other ST variants in estimating the true gradient.

**Classification with Deep SBN** In this section we verify Alg. 1 with different choice of noises in a deep network and verify optimization in binary weight probabilities using SGD on latent weights with Alg. 2. We consider CIFAR-10 dataset and use the SBN model illustrated in Fig. 4. The SBN model, its initialization and the full learning setup is detailed in Appendix D.3. We trained this SBN with three choices of noise distributions corresponding to proxies used by prior work as in Fig. 1 (c-e). Table 1 shows the test results in comparison with baselines.

We see that training with different choices of noise distributions, corresponding to different ST rules, all achieves similar results. This is in contrast to empirical studies advocating specific proxies and is allowed by the consistency of the model, initialization and training. The identity ST applied to weights, implementing SMD updates, works well. Comparing to empirical ST baselines (all except Peters & Welling), we see that there is no significant difference in the 'det' column indicating that our derived ST method is on par with the well-guessed baselines. If the same networks are tested in the stochastic mode ('10-sample' column), there is a clear boost of performance, indicating an advantage of SBN models. Out of the two experiments of Hubara et al., randomized training (rand.) also appears better confirming advantage of stochastic ST. In the stochastic mode, there is a small gap to Peters & Welling, who use a different estimation method and pretraining. Pretraining a real valued network also seem important, *e.g.*, [19] report 91.7% accuracy with VGG-Small using pretraining and a smooth transition from continuous to binarized model. When our method is applied with an initialization from a pretrained model, improved results (92.6% 10-sample test accuracy) can be obtained with even a smaller

**Table 1:** Test accuracy for different methods on CIFAR-10 with the same/similar architecture. SBN can be tested either with zero noises (*det*) or using an ensemble of several samples (we use *10-sample*). Standard deviations are given w.r.t. to 4 trials with random initialization. The two quotations for Hubara et al. [27] refer to their result with Torch7 implementation using randomized Htanh and Theano implementation using deterministic Htanh, respectively.

**STOCHASTIC TRAINING**

| Method | det | 10-sample |
|---|---|---|
| Our SBN, logistic noise | $89.6 \pm 0.1$ | $90.6 \pm 0.2$ |
| Our SBN, uniform noise | $89.7 \pm 0.2$ | $90.5 \pm 0.2$ |
| Our SBN, triangular noise | $89.5 \pm 0.2$ | $90.0 \pm 0.3$ |
| Hubara et al. [27] (rand.) | 89.85 | - |
| Peters & Welling [43] | 88.61 | 16-sample: |
|  |  | 91.2 |

**DETERMINISTIC TRAINING**

| | | |
|---|---|---|
| Rastegari et al. [45] | 89.83 | - |
| Hubara et al. [27] (det.) | 88.60 | - |

**Fig. 5:** Schematic explanation of the optimization process using a biased estimator followed by a correction with an unbiased estimator. Initially, the biased estimator makes good progress, but then the value of the true loss function may start growing while the optimization steps nevertheless come closer to the optimal location in the parameter space.

network [35]. There are however even more superior results in the literature, *e.g.*, using neural architecture search with residual real connections, advanced data augmentation techniques and model distillation [10] achieve 96.1%.

The takeaway message here is that ST can be considered in the context of deep SBN models as a simple and robust method if the estimator matches the model and is applied correctly. Since we achieve experimentally near 100% training accuracy in all cases, the optimization fully succeeds and thus the bias of ST is tolerable.

## 5    Conclusion

We have put many ST methods on a solid basis by deriving and explaining them from the first principles in one framework. It is well-defined what they estimate and what the bias means. We obtained two different main estimators for propagating activations and weights, bringing the understanding which function they have, what approximations they involve and what are the limitations imposed by these approximations. The resulting methods in all cases are strikingly simple, no wonder they have been first discovered empirically long ago. We showed how our theory leads to a useful understanding of bias properties and to reasonable choices that allow for a more reliable application of these methods. We hope that researchers will continue to use these simple techniques, now with less guesswork and obscurity, as well as develop improvements to them.

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

# Appendix

## A   Related Work

**Hinton's vs Bengio's ST**   The name *straight-through* and the first experimental comparison was proposed by Bengio et al. [5]. Referring to Hinton's lecture, they describe the idea as "*simply to back-propagate through the hard threshold function as if it had been the identity function*". In the aforementioned lecture [25], however we find a somewhat different description: "*during the forward pass we stochastically pick a binary value using the output of the logistic, and then during the backward pass we pretend that we've transmitted the real valued probability from the logistic*". We can make two observations: 1) different variants appeared early on and 2) many subsequent works [*e.g.* 58] attribute these two variants in the exact opposite way, adding to the confusion.

**ST Analysis**   Yin et al. [58] analyzes deterministic ST variants. The theoretical analysis is applicable to 1 hidden layer model with quadratic loss and the input data following a Gaussian distribution. The input distribution assumption is arguably artificial, however it allows to analyze the expected loss and its gradient. They show that population ST gradients using ReLU and clipped ReLU proxy correlate positively with the true population gradient and allow for convergence while identity ST does not. In Appendix B we show that in the SBN model, a simple correction of the quadratic loss function makes the base ST estimator unbiased and all rescaled estimators including identity are ascent directions in the expectation. Also note that the approach to analyze deterministic ST methods by considering the expectation over the input has a principle limitation for extending to deep models: the expectation over the input of a deterministic network with two hidden binary layers is still non-smooth (non-differentiable) in the parameters of the second layer.

Cheng et al. [13] shows for networks with 1 hidden layer that STE is approximately related to the projected Wasserstein gradient flow method proposed there.

On the weights side of the problem, Ajanthan et al. [1] connected mirror descent updates for constrained optimization (*e.g.*, $\boldsymbol{w} \in [0,1]^m$) with straight-through methods. The connection of deterministic straight-through for weights and proximal updates was also observed in [4]. Mirror Descent has been applied to variational Bayesian learning of continuous weights *e.g.* in Lin et al. [33], taking the form of update in natural parameters with the gradient in the mean parameters, same as in our case.

**Alternative Estimators**   For deep binary networks several gradient estimation approaches are based on stochastic gradients of analytically smoothed/approximated loss [43, 46]. There is however a discrepancy between analytic approximation and the binary samples used at the test time. Shekhovtsov et al. [48,

Fig. 4] show that such relaxed objectives may indeed significantly diverge during the training. To obtain good results, a strong dropout regularization and/or pretraining is needed [43, 46]. Despite these difficulties they demonstrate on par or improved results, especially when using average prediction over multiple noise samples at test time.

Dai et al. [17] perform a correct interchange of derivative and integral in (3) using *weak (distributional)* derivatives. After computing local expectations in this more complicated formalism, they are back with finite differences (6) which they also propose to linearize as in (7). Thus their *distributional SGD is equivalent to common SGD with the ST estimator* Alg. 1.

**Bayesian Learning** Bayesian learning in the form of simple update rules are recently (contemporaneously) studied by Khan & Rue [30]. We emphasize the interplay with the identity-ST estimator and the connection to the implicit regularization. The recent work by Meng et al. [37] proposed Bayesian learning with binary weights using Gumbel-Softmax estimator of gradients. We analyze it in [47] and demonstrate that it reduces to a different, non-Bayesian, rule when applied in large-scale experiments.

# B   Analysis of ST with 1 Hidden Layer

## B.1   Invariances

We have the following simple yet desirable and useful property. It is easy to observe that binary activations admit equivalent reformulations as

$$\text{sign}(a_i - z_i) = \text{sign}(T(a_i) - T(z_i)) \tag{16}$$

for any strictly monotone mapping $T \colon \mathbb{R} \to \mathbb{R}$.

**Proposition B.1.** *The gradient computed by Alg. 1 is invariant to equivalent transformations under* sign *as in* (16).

*Proof.* Let us denote the transformed noise as $\tilde{z}_i = T(z_i)$, its cdf as $G$ and the transformed activations as $\tilde{a}_i = T(a_i)$. The sampling probability in line 2 of Alg. 1 does not change since after the transformation it computes $p = G(\tilde{a}_i) = \mathbb{P}(\tilde{z}_i \leq \tilde{a}_i \mid \tilde{a}_i) = \mathbb{P}(z_i \leq a_i \mid a_i) = F(a_i)$. The gradient returned by line 5 does not change since we have $\frac{d}{da_i}G(T(a_i)) = F'(a_i)$.                              $\square$

In contrast, empirical straight-through approaches where the proxy is hand-designed fail to maintain this property. In particular, in the deterministic straight-through approach transforms such as $\text{sign}(a_i) = \text{sign}(T(a_i))$ while keeping the proxy of sign used in backprop fixed lead to different gradient estimates. This partially explains why many proxies have been tried, *e.g.* ApproxSign [34], and their scale needed tuning. Another pathological special case that leads to a confusion between identity straight-through and other forms is as follows.

**Corollary B.1.** *Let $F$ be strictly monotone. Then letting $T = F$ leads to $T(z_i)$ being uniformly distributed. Let $\tilde{a}_i = T(a_i)$. In this case the backpropagation rule*

*in line 5 of Alg. 1 can be interpreted as replacing the gradient of $\text{sign}(\tilde{a}_i - T(z_i))$ in $\tilde{a}_i$ with just identity.*

Indeed, since $\tilde{z}_i = T(z_i)$ is uniform, we have $G' = 1$ on $(0, 1)$ and $\tilde{a}_i = F(a_i)$ is guaranteed to be in $(0, 1)$ by strict monotonicity. The gradient back-propagated by usual rules through $\tilde{a}_i$ (outside of the ST Alg. 1) encounters derivative of $F$ as before. Hence we see that the description "to back-propagate through the hard threshold function as if it had been the identity function" could be misleading as the resulting estimator crucially depends on what transformations are applied under the hard threshold despite they do not affect the network predictions in any way. We refer to the variant by [5] as identity-ST, as it specifically uses the identity proxy for the gradient in the pre-sigmoid activation.

## B.2   Bias Analysis

**I)** Since the only approximation that we made was linearization of the objective $\mathcal{L}$, we have the following basic property.

**Proposition B.2.** *If the objective function $\mathcal{L}$ is multilinear[4] in the binary variables $\boldsymbol{x}$, then Alg. 1 is unbiased.*

*Proof.* In this case (7) holds as equality.                                      □

While extremely simple, this is an important point for understanding the ST estimator. As an immediate consequence we can easily design counter-examples where ST is wrong.

**Example 1.** Let $a \in \mathbb{R}$, $x = \text{sign}(a - z)$ and $\mathcal{L}(x) = x^2$. In this case the expected value of the loss is 1, independent of $a$. The true gradient is zero. However the expected ST gradient is $\mathbb{E}[2F'(a)2x] = 4F'(a)(2\mathcal{L}(a) - 1)$ and can be positive or negative depending on $a$.

**Example 2** (Tokui & Sato 54)**.** Let $\mathcal{L}(x) = x - \sin(2\pi x)$. Then the finite difference $\mathcal{L}(1) - \mathcal{L}(0) = 1$ but the derivative $\frac{\partial \mathcal{L}}{\partial x} = 1 - 2\pi \cos(2\pi x) = -1$. In this failure example, ST, even in the expectation, will point to exactly the opposite direction of the true gradient.

An important observation from the above examples is that the result of ST is not invariant with respect to reformulations of the loss that preserve its values in all binary points. In particular, we have that $\mathcal{L} \equiv 1$ in the first example and $\mathcal{L}(x) \equiv x$ in the second example for any $x \in \{-1, 1\}$. If we used these equivalent representations instead, the ST estimator would have been correct.

More generally, any real-valued function of binary variables has a unique polynomial (and hence multilinear) representation [7] and therefore it is possible to find a loss reformulation such that the ST estimator will be unbiased. Unfortunately, this representation is intractable in most cases, but it is tractable, *e.g.*, for a quadratic loss, useful in regression and autoencoders with a Gaussian observation model.

---

[4] E.g. $x_1 x_2 x_3$ is trilinear and thus qualifies but $x_1^2$ is not multi-linear.

**Proposition B.3.** *Let $\mathcal{L}(\boldsymbol{x}) = \|\boldsymbol{W}\boldsymbol{x} - \boldsymbol{y}\|^2$. Then the multilinear equivalent reformulation of $\mathcal{L}$ is given by*

$$\tilde{\mathcal{L}}(\boldsymbol{x}) = \|\boldsymbol{W}\boldsymbol{x} - \boldsymbol{y}\|^2 - \sum_i x_i^2 \|\boldsymbol{W}_{:,i}\|^2 + \sum_i \|\boldsymbol{W}_{:,i}\|^2, \tag{17}$$

*where $\boldsymbol{W}_{:,i}$ is the $i$'th column of $\boldsymbol{W}$.*

*Proof.* By expanding the square and using the identity $x_i^2 = 1$ for $x_i \in \{-1, 1\}$.
$\square$

Simply adjusting the loss using this equivalence and applying ST to it, fixes the bias problem.

**II)** Next we ask the question, whether dropping the multiplier $\mathrm{diag}(F'(\boldsymbol{a}))$ or changing it by another multiplier, which we call an *(internal) rescaling* of the estimator, can lead to an incorrect estimation.

**Proposition B.4.** *If instead of $\mathrm{diag}(F'(\boldsymbol{a}))$ any positive semidefinite diagonal matrix $\boldsymbol{\Lambda}$ is used in Alg. 1, the expected rescaled estimator preserves non-negative scalar product with the original estimator.*

*Proof.* We write the chain (9) in a matrix form as $\boldsymbol{J}_1^\mathsf{T} \boldsymbol{\Lambda}_0(\boldsymbol{a}) \boldsymbol{J}_2^\mathsf{T}(\boldsymbol{x})$, with the Jacobians $\boldsymbol{J}_1 = \frac{\partial \boldsymbol{a}}{\partial \boldsymbol{\phi}}$, $\boldsymbol{\Lambda}^0 = \mathrm{diag}(F'(\boldsymbol{a}))$ and $\boldsymbol{J}_2(\boldsymbol{x}) = \frac{\partial \mathcal{L}(\boldsymbol{x})}{\partial \boldsymbol{x}}$. The modified gradient with $\boldsymbol{\Lambda}$ is then defined as $\boldsymbol{J}_1^\mathsf{T} \boldsymbol{\Lambda}(\boldsymbol{a}) \boldsymbol{J}_2^\mathsf{T}(\boldsymbol{x})$.

We are interested in the scalar product between the expected gradient estimates:

$$\langle \mathbb{E}[\boldsymbol{J}_1^\mathsf{T} \boldsymbol{\Lambda}_0 \boldsymbol{J}_2^\mathsf{T}], \mathbb{E}[\boldsymbol{J}_1^\mathsf{T} \boldsymbol{\Lambda} \boldsymbol{J}_2^\mathsf{T}] \rangle, \tag{18}$$

where the expectation is over $\boldsymbol{x}$. Since neither $\boldsymbol{J}_1$ nor $\boldsymbol{\Lambda}$, $\boldsymbol{\Lambda}_0$ depend on $\boldsymbol{x}$, we can move the expectations to $\boldsymbol{J}_2$. Let $\bar{\boldsymbol{J}}_2 = \mathbb{E}\left[\frac{\partial \mathcal{L}(\boldsymbol{x})}{\partial \boldsymbol{x}}\right]$. Then the scalar product between the expected estimates becomes

$$\langle \boldsymbol{J}_1^\mathsf{T} \boldsymbol{\Lambda}_0 \bar{\boldsymbol{J}}_2^\mathsf{T}, \boldsymbol{J}_1^\mathsf{T} \boldsymbol{\Lambda} \bar{\boldsymbol{J}}_2^\mathsf{T} \rangle = \mathrm{Tr}(\bar{\boldsymbol{J}}_2 \boldsymbol{\Lambda} \boldsymbol{J}_1 \boldsymbol{J}_1^\mathsf{T} \boldsymbol{\Lambda}_0 \bar{\boldsymbol{J}}_2^\mathsf{T}). \tag{19}$$

Notice that $\boldsymbol{J}_1 \boldsymbol{J}_1^\mathsf{T}$ is positive semi-definite, $\boldsymbol{\Lambda}_0$ is also positive semi-definite since it is diagonal with non-negative entries. It follows that $\boldsymbol{R} = \boldsymbol{\Lambda} \boldsymbol{J}_1 \boldsymbol{J}_1^\mathsf{T} \boldsymbol{\Lambda}_0$ is positive semidefinite and that $\bar{\boldsymbol{J}}_2 \boldsymbol{R} \boldsymbol{J}_2^\mathsf{T}$ is positive semi-definite. Its trace is non-negative.
$\square$

We obtained that the use of an internal rescaling, in particular identity instead of $F'$, is not too destructive: if Alg. 1 was unbiased, the rescaled estimator may be biased but it is guaranteed to give an ascend direction in the expectation so that the optimization can in principle succeed. However, assuming that Alg. 1 is biased (when $\mathcal{L}$ is not multi-linear) but gives an ascent direction in the expectation, *the ascent direction property cannot be longer guaranteed for the rescaled gradient.*

**III)** Next, we study whether the ST gradient is a valid ascent direction even when $\mathcal{L}$ is not multi-linear.

**Proposition B.5.** *Let $\mathcal{L}(\boldsymbol{x})$ be such that its partial derivative $g_i = \frac{\partial \mathcal{L}}{\partial x_i}$ as a function of $x_i$ is Lipschitz continuous for all $i$ with a constant $L$. Then the expected ST gradient is an ascent direction for any $\boldsymbol{a}(\boldsymbol{\phi})$ and $\mathcal{L}(\boldsymbol{x})$ if and only if*

$$\left| \mathbb{E}[g_i] \right| > L \text{ for all } i. \tag{20}$$

*Proof. Sufficiency (if part).* The true gradient using the local expectation form (6) expresses as

$$\mathbb{E}\Big[\sum_i \big(\tfrac{\partial a_i}{\partial \boldsymbol{\phi}}\big)\big(p_z(a_i)\big)x_i\big(\mathcal{L}(\boldsymbol{x})-\mathcal{L}(\boldsymbol{x}_{\downarrow i})\big)\Big] = \mathbb{E}[J\Delta], \tag{21}$$

where the expectation is w.r.t. $\boldsymbol{x} \sim p(\boldsymbol{x};\boldsymbol{\phi})$ and we introduced the matrix notation $\boldsymbol{J} = \big(\tfrac{\partial \boldsymbol{a}}{\partial \boldsymbol{\phi}}\big)^{\mathsf{T}} \mathrm{diag}(p_z(\boldsymbol{a}))$, and $\Delta_i = x_i\big(\mathcal{L}(\boldsymbol{x})-\mathcal{L}(\boldsymbol{x}_{\downarrow i})\big)$. The ST gradient replaces $\Delta_i$ with $2g_i(\boldsymbol{x})$. Since in both cases $\boldsymbol{J}$ does not depend on $\boldsymbol{x}$, the expectation can be moved to the last term. Respectively, let us define $\bar{\boldsymbol{\Delta}} = \mathbb{E}[\boldsymbol{\Delta}]$ and $\bar{\boldsymbol{g}} = \mathbb{E}[\boldsymbol{g}]$. The scalar product between the true gradient and the expected ST gradient can then be expressed as

$$\langle \boldsymbol{J}\bar{\boldsymbol{\Delta}}, \boldsymbol{J}\bar{\boldsymbol{g}} \rangle = \mathrm{Tr}(\boldsymbol{J}\bar{\boldsymbol{g}}\bar{\boldsymbol{\Delta}}^{\mathsf{T}}\boldsymbol{J}^{\mathsf{T}}). \tag{22}$$

From the relation

$$x_i(\mathcal{L}(\boldsymbol{x}) - \mathcal{L}(\boldsymbol{x}_{\downarrow i})) = \int\limits_{-1}^{1} g_i(\boldsymbol{x})\mathrm{d}x_i \tag{23}$$

and Lipschitz continuity of $g_i$ in $x_i$ we have bounds

$$2(g_i(\boldsymbol{x}) - L) \le x_i(\mathcal{L}(\boldsymbol{x}) - \mathcal{L}(\boldsymbol{x}_{\downarrow i})) \le 2(g_i(\boldsymbol{x}) + L). \tag{24}$$

It follows that

$$2(\mathbb{E}[\boldsymbol{g}] - L) \le \mathbb{E}[\boldsymbol{\Delta}] \le 2(E[\boldsymbol{g}] + L), \tag{25}$$

coordinate-wise. The outer product $\bar{\boldsymbol{g}}\bar{\boldsymbol{\Delta}}^{\mathsf{T}}$ is positive semidefinite iff $\bar{g}_i\bar{\Delta}_i \ge 0$ for all $i$. According to bounds above, this holds true if

$$(\forall i \,|\, \bar{g}_i \ge 0) \quad 2(|\bar{g}_i| - L) \ge 0 \tag{26}$$

$$(\forall i \,|\, \bar{g}_i < 0) \quad 2(|\bar{g}_i| + L) \le 0, \tag{27}$$

or simply $(\forall i)\ |\bar{g}_i| \ge L$.

*Necessity (only if part).* We want to show that the requirements (20), which are simultaneous for all coordinates of $\boldsymbol{g}$, cannot be relaxed unless we make some further assumptions about $\boldsymbol{a}$ or $\mathcal{L}$. Namely, if $\exists i^*$ such that $\bar{g}_{i^*}\bar{\Delta}_{i^*} < 0$, then there exists $\boldsymbol{a}$ such that $\langle \boldsymbol{J}\bar{\boldsymbol{g}}, \boldsymbol{J}\bar{\boldsymbol{\Delta}} \rangle < 0$. *I.e.* a single wrong direction can potentially be rescaled by the downstream Jacobians to dominate the contribution of other components. This is detailed in the following steps.

Assume $(\exists i^*)\ |\bar{g}_{i^*}| < L$. Then exists $\mathcal{L}(\boldsymbol{x})$ such that the bounds (24) are tight (*e.g.* $\mathcal{L}(x) = x^2$) and therefore there will hold $\bar{g}_{i^*}\bar{\Delta}_{i^*} < 0$. Since $\boldsymbol{\Lambda} = \mathrm{diag}(p_z(\boldsymbol{a}))$ is positive semi-definite, $\boldsymbol{\Lambda}\bar{\boldsymbol{g}}\bar{\boldsymbol{\Delta}}^{\mathsf{T}}\boldsymbol{\Lambda}$ will preserve the non-positive sign of the component $(i^*, i^*)$. There exists $\boldsymbol{a}(\boldsymbol{\phi})$ such that $\tfrac{\partial \boldsymbol{a}}{\partial \boldsymbol{\phi}}$ scales down all coordinates $i \ne i^*$ and scales up $i^*$ such that the $\mathrm{Tr}(\boldsymbol{J}\bar{\boldsymbol{g}}\bar{\boldsymbol{\Delta}}^{\mathsf{T}}\boldsymbol{J}^{\mathsf{T}})$ is dominated by the entry $(i^*, i^*)$. The resulting scalar product between the expected gradient and the true gradient thus can be negative. $\qquad\square$

**IV)** Next we study, a typical use case when hidden binary variables are combined using a linear layer, initialized randomly. A typical initialization procedure would rescale the weights according to the size of the fan-in for each output.

**Proposition B.6.** *Assume that the loss function is applied after a linear normalized transform of Bernoulli variables,* i.e., *takes the form*

$$\mathcal{L}(\boldsymbol{x}) = \ell(\boldsymbol{W}\boldsymbol{x}), \tag{28}$$

*where* $\boldsymbol{W} \in \mathbb{R}^{K \times n}$ *is a matrix of normally distributed weights, normalized to satisfy* $\|W_{k,:}\|_2^2 = 1 \; \forall k$. *Then the expected Lipschitz constant of gradients of* $\mathcal{L}$ *scales as* $O(\frac{1}{\sqrt{n}})$.

*Proof.* Let $\boldsymbol{u} = \boldsymbol{W}\boldsymbol{x}$ and let $\frac{\partial \ell}{\partial \boldsymbol{u}}$ be Lipschitz continuous with constant $L$. The gradient of $\mathcal{L}$ expresses as

$$g_i = \frac{\mathrm{d}\mathcal{L}(\boldsymbol{x})}{\mathrm{d}x_i} = \langle \frac{\partial \ell(\boldsymbol{u})}{\partial \boldsymbol{u}}, \boldsymbol{W}_{:,i} \rangle. \tag{29}$$

By assumptions of random initialization and normalization, $W_{k,i} \sim \mathcal{N}(0, \frac{1}{n})$. If we consider $|g_i|$ in the expectation over initialization we obtain that

$$\mathbb{E}_{\boldsymbol{W}}\big[|g_i(\boldsymbol{x}) - g_i(\boldsymbol{y})|\big] = \mathbb{E}_{\boldsymbol{W}}\big[\langle \ell'(\boldsymbol{W}\boldsymbol{x}) - \ell'(\boldsymbol{W}\boldsymbol{y}), \boldsymbol{W}_{:,i}\rangle\big] \leq L\mathbb{E}_{\boldsymbol{W}}\big[\|\boldsymbol{W}_{:,i}\|\big] = LK\sqrt{\tfrac{2}{n\pi}}. \tag{30}$$

Therefore $g_i$ has expected Lipschitz constant $LK\sqrt{\frac{2}{n\pi}}$. □

The normal distribution assumption is not principal for conclusion of $O(\frac{1}{\sqrt{n}})$ dependance. Indeed, for any distribution with a finite variance it would hold as well, differing only in the constant factors. We obtain an important corollary.

**Corollary B.2.** *As we increase the number of hidden binary units* $n$ *in the model, the bias of ST decreases, at least at initialization.*

**V)** Finally, we study conditions when a deterministic version of ST gives a valid ascent direction.

**Proposition B.7.** *Let* $\boldsymbol{x}^* = \mathrm{sign}(\boldsymbol{a})$. *Let* $g_i = \frac{\partial \mathcal{L}(\boldsymbol{x})}{\partial x_i}$ *be Lipschitz continuous with constant L. Let* $\boldsymbol{g}^* = \boldsymbol{g}(\boldsymbol{x}^*)$ *and* $p^* = p(\boldsymbol{x}^*|\boldsymbol{a})$. *The deterministic ST gradient at* $\boldsymbol{x}^*$ *forms a positive scalar product with the expected stochastic ST gradient if*

$$|g_i^*| \geq 2(1 - p^*)L \quad \forall i. \tag{31}$$

*Proof.* Similarly to Proposition B.5, let $\boldsymbol{J} = \left(\frac{\partial \boldsymbol{a}}{\partial \boldsymbol{\phi}}\right)^{\mathsf{T}} \mathrm{diag}(p_z(\boldsymbol{a}))$. The scalar product between the expected ST gradient and the deterministic ST gradient is given by

$$\langle \boldsymbol{J}\mathbb{E}[\boldsymbol{g}(\boldsymbol{x})], \boldsymbol{J}\boldsymbol{g}^* \rangle = \mathrm{Tr}\left(\mathbb{E}[\boldsymbol{g}(\boldsymbol{x})]\boldsymbol{g}^{*\mathsf{T}}\boldsymbol{J}^{\mathsf{T}}\right). \tag{32}$$

In order for it to be non-negative we need $\mathbb{E}[g(\boldsymbol{x})_i]g_i^* \geq 0 \; \forall i$. Observe that $\mathbb{E}[g(\boldsymbol{x})_i]$ is a sum that includes $g_i^*$ with the weight $p^*$. We therefore need

$$\sum_{\boldsymbol{x} \neq \boldsymbol{x}^*} p(\boldsymbol{x}|\boldsymbol{a})g(\boldsymbol{x})_i g_i^* + p^* g_i^{*2} \geq 0. \tag{33}$$

From Lipschitz continuity of $g_i$ we have the bound $|g(\boldsymbol{x})_i - g_i^*| \leq L|x_i - x_i^*|$, or using that $|x_i - x_i^*| \leq 2$ we have

$$g_i^* - 2L \leq g(\boldsymbol{x})_i \leq g_i^* + 2L. \tag{34}$$

Therefore

$$g(\boldsymbol{x})_i g_i^* \geq g_i^{*2} - 2L|g_i^*|. \tag{35}$$

We thus can lower bound (33) as

$$\sum_{\boldsymbol{x} \neq \boldsymbol{x}^*} p(\boldsymbol{x}|\boldsymbol{a})(|g_i^*| - 2L)|g_i^*| + p^* g_i^{*2} = -2L|g_i^*|(1 - p^*) + g_i^{*2}. \tag{36}$$

This lower bound is non-negative if

$$|g_i^*| \geq 2L(1 - p^*). \tag{37}$$

$\square$

Compared to Proposition B.5, this condition has an extra factor of $2(1 - p^*)$. Since $p^*$ is the product of probabilities of all units $x_i^*$, we expect initially $p^* \ll 1$. This condition improves at the same rate with the increase in the number of hidden units as the case covered by Proposition B.6. In addition it becomes progressively more accurate as units learn to be more deterministic, because in this case the factor $(1 - p^*)$ decreases. However, note that this proposition describes the gap between deterministic ST and stochastic ST. And even when this gap diminishes, the gap between ST and the true gradient remains.

We can obtain a similar sufficient condition for the scalar product between deterministic ST and the executed true gradient, that (unlike the direct combination of Proposition B.5 and Proposition B.7) ensures an ascent direction.

**Proposition B.8.** *Let $\boldsymbol{x}^* = \mathrm{sign}(\boldsymbol{a})$. Let $g(\boldsymbol{x})_i = \frac{\partial \mathcal{L}(\boldsymbol{x})}{\partial x_i}$ be Lipschitz continuous with constant $L$. Let $\boldsymbol{g}^* = \boldsymbol{g}(\boldsymbol{x}^*)$ and $p^* = p(\boldsymbol{x}^*|\boldsymbol{a})$. The deterministic ST gradient at $\boldsymbol{x}^*$ forms a positive scalar product with the true gradient if*

$$|g_i^*| \geq 2(1 - p^*)L + L \quad \forall i. \tag{38}$$

*Proof.* The proof is similar to Proposition B.7, only in this case we need to ensure $\mathbb{E}[\Delta_i]g_i^* \geq 0$. Using (25) we get the bounds

$$2(\mathbb{E}[\boldsymbol{g}] - L) \leq \mathbb{E}[\boldsymbol{\Delta}] \leq 2(E[\boldsymbol{g}] + L), \tag{39}$$

And using additionally (34) we get

$$2(p^* g_i^* + (1 - p^*)(g_i^* - 2L) - L) \leq \mathbb{E}[\Delta_i] \leq 2(p^* g_i^* + (1 - p^*)(g_i^* + 2L) + L). \tag{40}$$

Collecting the terms

$$2(g_i^* - (1 - p^*)2L - L) \leq \mathbb{E}[\Delta_i] \leq 2(g_i^* + (1 - p^*)2L + L). \tag{41}$$

Multiplying by $g_i^*$ we obtain that a sufficient condition for $\mathbb{E}[\Delta_i]g_i^* \geq 0$ is

$$|g_i^*| \geq (1 - p^*)2L + L. \tag{42}$$

$\square$

## C   Mirror Descent and Variational Mirror Descent

### C.1   Mirror Descent

Mirror descent is a widely used method for constrained optimization of the form $\min_{\boldsymbol{x} \in \mathcal{X}} f(\boldsymbol{x})$, where $\mathcal{X} \subset \mathbb{R}^n$, introduced by Nemirovsky & Yudin [39]. Let $\Phi : \mathcal{X} \to \mathbb{R}$ be strictly convex and differentiable on $\mathcal{X}$, called a *mirror map*. *Bregman divergence* $D_\Phi(\boldsymbol{x}, \boldsymbol{y})$ associated with $\Phi$ is defined as

$$D_\Phi(\boldsymbol{x}, \boldsymbol{y}) = \Phi(\boldsymbol{x}) - \Phi(\boldsymbol{y}) - \langle \nabla \Phi(\boldsymbol{y}), \boldsymbol{x} - \boldsymbol{y} \rangle. \tag{43}$$

An update of MD algorithm can be written as:

$$\boldsymbol{x}^{t+1} = \underset{\boldsymbol{x} \in \mathcal{X}}{\arg\min} \langle \boldsymbol{x}, \nabla f(\boldsymbol{x}^t) \rangle + \frac{1}{\varepsilon} D_\Phi(\boldsymbol{x}, \boldsymbol{x}^t). \tag{44}$$

In the unconstrained case when $\mathcal{X} = \mathbb{R}^n$ or in the case when the critical point is guaranteed to be in $\mathcal{X}$ (as typically ensured by the design of $D_\Phi$), the solution can be found from the critical point equations, leading to the general form of iterates

$$\nabla \Phi(\boldsymbol{x}^{t+1}) = \nabla \Phi(\boldsymbol{x}^t) - \varepsilon \nabla f(\boldsymbol{x}^t) \tag{45}$$
$$\boldsymbol{x}^{t+1} = (\nabla \Phi)^{-1} \left( \nabla \Phi(\boldsymbol{x}^t) - \varepsilon \nabla f(\boldsymbol{x}^t) \right).$$

**Proposition 1.** *Common SGD in latent weights $\boldsymbol{\eta}$ using the* identity straight-through-weights *Alg. 2 implements SMD in the weight probabilities $\boldsymbol{\theta}$ with the divergence corresponding to $F$.*

*Proof.* The proof closely follows Ajanthan et al. [1]. Differently from us, they considered deterministic ST. Their argumentation includes taking the limit in which $F$ is squashed into the step function and which renders MD invalid. This limit is not needed in our formulation.

We start from the defining equation of MD update in the form (45). In order for (45) to match common SGD on $\boldsymbol{\eta}$ with $\eta_i = F^{-1}(\theta_i)$, the mirror map $\Phi$ must satisfy $\nabla \Phi(\boldsymbol{\theta}) = F^{-1}(\boldsymbol{\theta})$, where $F^{-1}$ is coordinate-wise. We can therefore consider coordinate-wise mirror maps $\Phi \colon \mathbb{R} \to \mathbb{R}$. The inverse $F^{-1}$ exists if $F$ is strictly monotone, meaning that the noise density is non-zero on the support. Finding the mirror map $\Phi$ explicitly is not necessary for our purpose, however in 1D case it can be expressed simply as $\Phi(x) = \int_0^x F^{-1}(\eta) d\eta$. With this coordinate-wise mirror map, the MD update can be written as

$$\boldsymbol{\eta}^{t+1} = \boldsymbol{\eta}^t - \varepsilon \frac{d\mathcal{L}}{d\boldsymbol{\theta}} \bigg|_{\boldsymbol{\theta} = F(\boldsymbol{\eta}^t)}. \tag{46}$$

Thus MD on $\boldsymbol{\theta}$ takes the form of a descent step on $\boldsymbol{\eta}$ with the gradient $\frac{d\mathcal{L}}{d\boldsymbol{\theta}}$. A common SGD on $\boldsymbol{\eta}$ would use the gradient $\frac{d\mathcal{L}}{d\boldsymbol{\eta}} = \frac{\partial \boldsymbol{\theta}}{\partial \boldsymbol{\eta}} \frac{\partial \mathcal{L}}{\partial \boldsymbol{\theta}}$. Thus (46) bypasses the Jacobian $\frac{\partial \boldsymbol{\theta}}{\partial \boldsymbol{\eta}}$. This is exactly what Alg. 2 does. More precisely, when applying

the same derivations that we used to obtain ST for activations in order to esti-mate $\frac{d\mathcal{L}}{d\boldsymbol{\theta}}$, since $F(\eta_i) = \theta_i$, we obtain that the factor $\frac{\partial}{\partial\boldsymbol{\theta}}p(w_i;\theta)$, present in (6), expresses as

$$\frac{dF(\boldsymbol{\eta})}{d\boldsymbol{\theta}} = \frac{\partial F(F^{-1}(\boldsymbol{\theta}))}{\partial\boldsymbol{\theta}} = 1 \tag{47}$$

and thus can be omitted from the chain rule as defined in Alg. 2.               $\square$

## C.2 Latent Weight Decay Implements Variational Bayesian Learning

In the Bayesian learning setting we consider a model with binary weights $\boldsymbol{w}$ and are interested in estimating $p(\boldsymbol{w}|D)$, the posterior distribution of the weights given the data $D$ and the weights prior $p(\boldsymbol{w})$. In the variational Bayesian (VB) formulation, this difficult and multi-modal posterior is approximated by a sim-pler one $q(\boldsymbol{w})$, commonly a fully factorized distribution. The approximation is achieved by minimizing $\mathrm{KL}(q(\boldsymbol{w})\|p(\boldsymbol{w}|D))$. Let $q(\boldsymbol{w}) = \mathrm{Ber}(\boldsymbol{w};\boldsymbol{\theta})$ and $p(\boldsymbol{w}) = \mathrm{Ber}(\boldsymbol{w};\frac{1}{2})$, both meant component-wise, i.e. fully factorized. Then the VB prob-lem takes the form

$$\arg\min_{\boldsymbol{\theta}} \left\{ -\mathbb{E}_{(\boldsymbol{x}^0,\boldsymbol{y})\sim\mathrm{data}}\left[\mathbb{E}_{\boldsymbol{w}\sim\mathrm{Ber}(\boldsymbol{\theta})}\left[\log p(\boldsymbol{y}|\boldsymbol{x}^0;\boldsymbol{w})\right]\right] + \tfrac{1}{N}\mathrm{KL}(\mathrm{Ber}(\boldsymbol{\theta})\|\mathrm{Ber}(\tfrac{1}{2})) \right\}, \tag{48}$$

where we have rewritten the data likelihood as expectation and hence the coef-ficient $1/N$ in front of the KL term appeared. This problem is commonly solved by SGD taking one sample from the training data and one sample of $\boldsymbol{w}$ and applying backpropagation [21]. We can in principle do the same by applying an estimator for the gradient in $\boldsymbol{\theta}$.

The trick that we apply, different from common practices, is not to compute the gradient of the KL term but to keep this term explicit throughout to the proximal step leading to a *composite* MD [59]. With this we have

**Proposition 2.** *Common SGD in latent weights $\boldsymbol{\eta}$ with a weight decay and identity straight-through-weights Alg. 2 is equivalent to optimizing a factorized variational approximation to the weight posterior $p(\boldsymbol{w}|data)$ using a composite SMD method.*

*Proof.* Expanding data log-likelihood as the sum over all data points, we get

$$\log p(D\,|\,\boldsymbol{w}) = \sum_i \log p(x_i\,|\,\boldsymbol{w}) =: \sum_i l_i(\boldsymbol{w}). \tag{49}$$

When multiplying with $\frac{1}{N}$, the first term becomes the usual expected data like-lihood, where the expectation is in training data and weights $\boldsymbol{w} \sim q(\boldsymbol{w})$. Ex-panding also the parametrization of $q(\boldsymbol{w}) = \mathrm{Ber}(\boldsymbol{w}\,|\,\boldsymbol{\theta})$, the variational inference reads

$$\arg\min_{\boldsymbol{\theta}} \left\{ -\mathbb{E}_{\boldsymbol{w}\sim\mathrm{Ber}(\boldsymbol{\theta})}\left[\tfrac{1}{N}\sum_i l_i(\boldsymbol{w})\right] + \tfrac{1}{N}\mathrm{KL}(q(\boldsymbol{w})\|p(\boldsymbol{w})) + const \right\}. \tag{50}$$

We employ mirror descent to handle constraints $\boldsymbol{\theta} \in [0,1]^m$ similar to the above but now we apply it to this composite function, linearizing only the data part and keeping the prior KL part non-linear. Let

$$\boldsymbol{g} = \frac{1}{|I|} \sum_{i \in I} \nabla_{\boldsymbol{\theta}} \mathbb{E}_{\boldsymbol{w} \sim \mathrm{Ber}(\boldsymbol{\theta})} l_i(\boldsymbol{w})$$

be the stochastic gradient of the data term in the weight probabilities $\boldsymbol{\theta}$ using a min-batch $I$. The SMD step subproblem reads

$$\min_{\boldsymbol{\theta}} \left\{ \boldsymbol{g}^{\mathsf{T}}\boldsymbol{\theta} + \tfrac{1}{\varepsilon}\mathrm{KL}(\mathrm{Ber}(\boldsymbol{\theta})\|\mathrm{Ber}(\boldsymbol{\theta}^t)) + \tfrac{1}{N}\mathrm{KL}(\mathrm{Ber}(\boldsymbol{\theta})\|\mathrm{Ber}(\tfrac{1}{2})) \right\}. \tag{51}$$

We notice that $\mathrm{KL}(\mathrm{Ber}(\boldsymbol{\theta})\|\mathrm{Ber}(\tfrac{1}{2})) = -H(\mathrm{Ber}(\boldsymbol{\theta}))$, the negative entropy, and also introduce the prior scaling coefficient $\lambda = \frac{1}{N}$ in front of the entropy, which may optionally be lowered to decrease the regularization effect. With these notations, the composite proximal problem becomes

$$\min_{\boldsymbol{\theta}} \left\{ \boldsymbol{g}^{\mathsf{T}}\boldsymbol{\theta} + \tfrac{1}{\varepsilon}\mathrm{KL}(\mathrm{Ber}(\boldsymbol{\theta})\|\mathrm{Ber}(\boldsymbol{\theta}^t)) - \lambda H(\mathrm{Ber}(\boldsymbol{\theta})) \right\}. \tag{52}$$

The solution is found from the critical point equation in $\boldsymbol{\theta}$:

$$\nabla_{\boldsymbol{\theta}}\left(\boldsymbol{g}^{\mathsf{T}}\boldsymbol{\theta} + \tfrac{1}{\varepsilon}\mathrm{KL}(\mathrm{Ber}(\boldsymbol{\theta})\|\mathrm{Ber}(\boldsymbol{\theta}^t)) - \lambda H(\mathrm{Ber}(\boldsymbol{\theta}))\right) = 0 \tag{53a}$$

$$g_i + \tfrac{1}{\varepsilon}\left(\log \tfrac{\theta_i}{1-\theta_i} - \log \tfrac{\theta_i^t}{1-\theta_i^t}\right) + \lambda \log \tfrac{\theta_i}{1-\theta_i} = 0 \tag{53b}$$

$$(\varepsilon\lambda + 1)\log \tfrac{\theta_i}{1-\theta_i} = \log \tfrac{\theta_i^t}{1-\theta_i^t} - \varepsilon g_i \tag{53c}$$

$$\log \tfrac{\theta_i}{1-\theta_i} = \tfrac{1}{\varepsilon\lambda+1}\log \tfrac{\theta_i^t}{1-\theta_i^t} - \tfrac{\varepsilon}{\varepsilon\lambda+1}g_i. \tag{53d}$$

For the natural parameters we obtain:

$$\boldsymbol{\eta} = \tfrac{\boldsymbol{\eta}^t - \varepsilon\boldsymbol{g}}{\varepsilon\lambda+1} = \boldsymbol{\eta}^t - \tfrac{\varepsilon}{\varepsilon\lambda+1}\left(\lambda\boldsymbol{\eta}^t + \boldsymbol{g}\right). \tag{54}$$

We can further drop the correction of the step size $\frac{\varepsilon}{\varepsilon\lambda+1}$ since $\varepsilon\lambda + 1 \approx 1$ and the step size will need to be selected by cross validation anyhow. This gives us an update of the form

$$\boldsymbol{\eta} = \boldsymbol{\eta}^t - \varepsilon(\boldsymbol{g} + \lambda\boldsymbol{\eta}^t), \tag{55}$$

which is in the form of a standard step in any SGD or adaptive SGD optimizer. The difference is that the gradient in probabilities $\boldsymbol{\theta}$ is applied to make step in logits $\boldsymbol{\eta}$ and the prior KL divergence contributes the *logit decay* $\lambda$, which in this case is the *latent weight decay*. Since the ST gradient in $\boldsymbol{\theta}$ differs from the ST gradient in $\boldsymbol{\eta}$ by the factor $\mathrm{diag}(F')$, the claim of Proposition 2 follows. □

## D    Details of Experiments

### D.1    MNIST VAE

Here we give a specification of the experiment in Fig. 2, which illustrates the point that mismatching the constant factor in front of ST estimator leads to poor performance when the gradient is to be combined with other gradients, in this case with the analytic gradient of KL divergence in VAE.

**Dataset** We use *MNIST* data set[5]. It contains 60000 training and 10000 test images of handwritten digits. We used 50000 images for trainig, the reminder was kept as a validation set, however not utilized in this experiment.

**Preprocessing** No preprocessing or augmentation was performed. The grayscale image intensities in $[0,1]$ are interpreted as target Bernoulli probabilities for the decoder.

**Model** We used $\{0,1\}$ encoding of hidden states $x$. Closely following experiment design of [22, 42], we used the following network as encoder:

$$\text{Linear}(784,200) \to \tanh \to \text{Linear}(200,200) \to \tanh \to \text{Linear}(200,200).$$

The output of the encoder defines logits $\eta$ of the encoder Bernoulli model $p(x_i{=}1|\boldsymbol{y}) = \sigma(\eta_i)$. The decoder has the reverse architecture:

$$\text{Linear}(200,200) \to \tanh \to \text{Linear}(200,200) \to \tanh \to \text{Linear}(200,784)$$

and outputs logits $\boldsymbol{\nu}$ of conditionally independent Bernoulli generative model $p^{\text{dec}}(y_i{=}1|\boldsymbol{x}) = \sigma(\nu_i)$. The data images $\boldsymbol{t} \in \mathbb{R}^{784}$ are interpreted as target probabilities, and the negative conditional log-likelihood becomes

$$H = -\sum_i \left( t_i \log p^{\text{dec}}(y_i{=}1|\boldsymbol{x}) + (1 - t_i) \log p^{\text{dec}}(y_i{=}0|\boldsymbol{x}) \right). \tag{56}$$

We optimize the negative lower bound on the log-likelihood:

$$\mathcal{L} = \mathbb{E}_{\boldsymbol{y}\sim\text{data}}\Big[\mathbb{E}_{\boldsymbol{x}\sim p(\boldsymbol{x}|\boldsymbol{y})}\big[H\big] + \text{KL}(p(\boldsymbol{x}|\boldsymbol{y})\|p(\boldsymbol{x}))\Big], \tag{57}$$

where $p(\boldsymbol{x})$ is the uniform prior: $p(x_i) = \frac{1}{2}$.

**Optimization** We compute the KL term analytically for a mini-batch and use its exact gradient. The gradient of the expectation of $H$ is estimated. We used Adam optimizer with a learning rate in $\{0.001, 0.0003, 0.0001\}$.

### D.2   Stochastic Autoencoder

It was shown in the literature that semantic hashing using binary hash codes can achieve superior results using learned hash codes, in particular based on variational autoencoder (VAE) formulation, *e.g.*, recent works [12, 16, 38].

We propose a series of experiments that targets measuring the accuracy of gradient estimators through Bernoulli units and studying the dependence of this accuracy on the number of hidden units. It is appropriate to study here the plain stochastic autoencoder (2) and not a variational autoencoder (57)

---

[5] http://yann.lecun.com/exdb/mnist/

for the following reasons: 1) the gradient of prior KL term is known and need not be estimated, 2) VAE usually finds solutions in a partial posterior collapse (efficiently selecting the number of hidden units to use) which is in contradiction with our goal to study the dependence on the number of hidden units. In practice, the KL prior often needs to be tuned (in the public implementation of Ñanculef et al. [38] one can find $\beta = 0.015$ is used), which is complicating and irrelevant for our goals.

**Dataset**  The *20Newsgroups* data set[6] is a collection of approximately 20,000 text documents, partitioned (nearly) evenly across 20 different newsgroups. In our experiments we do not use the partitioning. We used the processed version of the dataset denoted as Matlab/Octave on the dataset's web site. It contains bag-of-words representations of documents given by one sparse word-document count matrix. We worked with the training set that contains 11269 documents in the bag of words representation.

**Preprocessing**  We keep only the 10000 most frequent words in the training set to reduce the computation requirements. Each of the omitted rare words occurs not more than in 10 documents.

**Reconstruction Loss**  Let $\boldsymbol{y} \in \mathbb{N}^d$ be the vector of word counts of a document and $\boldsymbol{x} \in \{0,1\}^n$ be a latent binary code representing the topic that we will learn. The decoder network given the code $\boldsymbol{x}$ deterministically outputs word frequencies $\boldsymbol{f} \in [0,1]^d$, $\sum_i f_i = 1$ and the reconstruction loss $-\log p^{\mathrm{dec}}(\boldsymbol{y}|\boldsymbol{x};\boldsymbol{\theta})$ is defined as

$$-\sum_i y_i \log f_i, \tag{58}$$

*i.e.*, the negative log likelihood of a generative model, where word counts $\boldsymbol{y}$ follow multinomial distribution with probabilities $\boldsymbol{f}$ and the number of trials equal to the length of the document. The encoder $p(\boldsymbol{x}|\boldsymbol{f};\boldsymbol{\phi})$ obtains word frequencies form $\boldsymbol{y}$ and maps them deterministically to Bernoulli probabilities $p(x_i|\boldsymbol{f};\boldsymbol{\phi})$. The loss of the autoencoder (2) is then

$$\mathbb{E}_{\boldsymbol{y}\sim\mathrm{data}}\big[\mathbb{E}_{\boldsymbol{z}\sim p(\boldsymbol{x}|\boldsymbol{y})}\big[-\log p^{\mathrm{dec}}(\boldsymbol{y}|\boldsymbol{x};\boldsymbol{\theta})\big]\big]. \tag{59}$$

**Networks**  The encoder network takes on the input word frequencies $\boldsymbol{f} \in \mathbb{R}^d$ and applies the following stack: FC($d \times 512$), ReLU, FC($512 \times n$), where FC is a fully connected layer. The output is the vector of logits of Bernoulli latent bits. The decoder network is symmetric: FC($n \times 512$), ReLU, FC($512 \times d$), Softmax. Its input is a binary latent code $\boldsymbol{x}$ and output is the word probabilities $\boldsymbol{f}$. Standard weight initialization is applied to all linear layers $\boldsymbol{W}$ setting $W_{i,j} \sim \mathcal{U}[-1/\sqrt{k}, 1/\sqrt{k}]$, where $k$ is the number of input dimensions to the layer. This is a standard initialization scheme [23], which is consistent with the assumptions we make in Proposition B.6 and hence important for verification of our analysis.

---

[6] http://qwone.com/~jason/20Newsgroups/

**Table D.1:**  List of estimators evaluated in the stochastic autoencoder experiment.

| Name | Details |
|---|---|
| ARM | State-of-the-art unbiased estimator [57]. |
| Gumbel($\tau$) | Gumbel-Softmax estimator [29] with temperature parameter $\tau$. |
| ST | Straight-Through Alg. 1. |
| det_ST | Deterministic version of ST setting the noise $\boldsymbol{z} = 0$ during training. |
| identity_ST | Identity ST variant described by [5]. |

**Estimators**  Estimators evaluated in this experiment are described in Table D.1. As detailed in Section 2, in the identity ST we still draw random samples in the forward pass like in Alg. 1 but omit the multiplication by $F'$. Alg. 1 is correctly instantiated for the $\{0, 1\}$ rather than $\pm 1$ encoding in all cases. For the ARM-10 correction phase and ARM-1000 ground truth estimation, the average of ARM estimates with the respective number of samples is taken.

**Optimizer**  We used Adam [31] optimizer with a fixed starting learning rate $lr = 0.001$ in both phases of the training. When switching to the ARM-10 correction phase, we reinitialize Adam in order to reset the running averages.

**Evaluation**  For each bit length we save the encoder and decoder parameter vectors $\boldsymbol{\phi}, \boldsymbol{\theta}$ every 100 epochs along the ARM training trajectory. At each such point, offline to the training, we first apply ARM-1000 in order to obtain an accurate estimate of the true gradient $\boldsymbol{g}$. We then evaluate each of the 1-sample estimators, including ARM itself ($=$ ARM-1).

The next question we discuss is how to measure the estimator accuracy. Clearly, if we just consider the expected local performance such as $\mathbb{E}[\langle \boldsymbol{g}, \tilde{\boldsymbol{g}} \rangle]$, unbiased estimators win regardless of their variance. This is therefore not appropriate for measuring their utility in optimization. We evaluate three metrics tailored for comparison of biased and unbiased estimators.

**Cosine Similarity**  This metric evaluates the expected cosine similarity, measuring alignment of directions:

$$\mathbb{E}\big[\langle \boldsymbol{g}, \tilde{\boldsymbol{g}} \rangle / (\|\boldsymbol{g}\|\|\tilde{\boldsymbol{g}}\|)\big], \tag{60}$$

where the expectation is over all training data batches and 100 stochastic trials of the estimator $\tilde{\boldsymbol{g}}$. This metric is well aligned with our theoretical analysis Section 2.1. It is however does not measure how well the gradient length is estimated. If the length has a high variance, this may hinder the optimization but would not be reflected by this metric.

**Expected Improvement**  To estimate the utility of the estimator for optimization, we propose to measure the expected optimization improvement using

the same proximal problem objective that is used in SGD or SMD to find an optimization step. Namely, let $\boldsymbol{g} = \nabla_\phi \mathcal{L}(\boldsymbol{\phi}^t)$ be the true gradient at the current point. Common SGD step is defined as

$$\boldsymbol{\phi}^{t+1} = \boldsymbol{\phi}^t + \arg\min_{\Delta\boldsymbol{\phi}} \left( \langle g, \Delta\boldsymbol{\phi} \rangle + \tfrac{1}{2\varepsilon} \|\Delta\boldsymbol{\phi}\|^2 \right). \tag{61}$$

The optimal solution is given by $\Delta\boldsymbol{\phi} = -\varepsilon\boldsymbol{g}$. Since instead of $\boldsymbol{g}$, only an approximation is available to the optimizer, we allow it to use the solution $\Delta\boldsymbol{\phi} = -\alpha\hat{\boldsymbol{g}}$, where $\hat{\boldsymbol{g}}$ is an estimator of $\boldsymbol{g}$ and $\alpha$ is one scalar parameter to adopt the step size. We then consider the expected decrease of the proxy objectives:

$$\mathbb{E}\left[ \langle \boldsymbol{g}, -\alpha\hat{\boldsymbol{g}} \rangle + \tfrac{\alpha^2}{2\varepsilon} \|\hat{\boldsymbol{g}}\|^2 \right]. \tag{62}$$

The parameter $\alpha$ correspond to a learning rate that can be tuned or adapted during learning. We set it optimistically for each estimator by minimizing the expected objective (62), which is a simple quadratic function in $\alpha$. One scalar $\alpha$ is thus estimated for one measuring point (*i.e.* for one expectation over all training batches and all 100 trials). As such, it is not overfitting to each estimator. The optimal $\alpha$ is given by

$$\alpha = \varepsilon\mathbb{E}[\langle \boldsymbol{g}, \hat{\boldsymbol{g}} \rangle]/\mathbb{E}[\|\hat{\boldsymbol{g}}\|^2] \tag{63}$$

and the value of the objective for this optimal $\alpha$ is

$$-\tfrac{\varepsilon}{2}\mathbb{E}[\langle \boldsymbol{g}, \hat{\boldsymbol{g}} \rangle]^2/\mathbb{E}[\|\hat{\boldsymbol{g}}\|^2]. \tag{64}$$

For the purpose of comparing estimators, $-\tfrac{\varepsilon}{2}$ is irrelevant and the comparison can be made on the square root of (64). We obtain an equivalent metric that is the expected loss decrease normalized by the RMS of the gradients:

$$-\mathbb{E}[\langle \boldsymbol{g}, \hat{\boldsymbol{g}} \rangle]/\sqrt{\mathbb{E}[\|\hat{\boldsymbol{g}}\|^2]}. \tag{65}$$

Confer to common adaptive methods which divide the step-length exactly by the square root of a running average of second moment of gradients, in particular Adam (applied per-coordinate there). This suggests that this metric is more tailored to measure the utility of the estimator for optimization. For brevity, we refer to (65) as *expected improvement*. Note also that in (65) we preserve the sign of $\mathbb{E}[\langle \boldsymbol{g}, \hat{\boldsymbol{g}} \rangle]$ and if the estimator is systematically in the wrong direction, we expect to measure a positive value in (65), *i.e.* predicting objective ascent rather than descent.

**Root Mean Squared Error** It is rather common to measure the error of biased estimators as

$$\text{RMSE} = \sqrt{\mathbb{E}[\|\boldsymbol{g} - \hat{\boldsymbol{g}}\|^2]}. \tag{66}$$

This metric however may be less indicative and less discriminative of the utility of the estimator for optimization. In Fig. D.1 it is seen that RMSE of ARM estimator can be rather high, especially with more latent bits, yet it performs rather well in optimization.

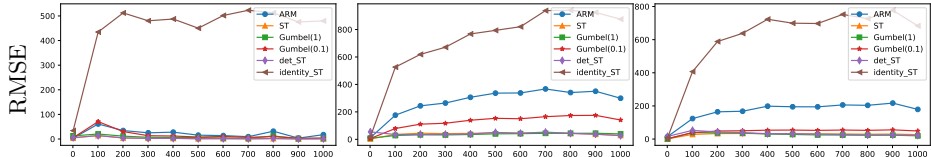

**Fig. D.1:** Root Mean Squared error of different estimators for the same reference trajectories as Fig. 3.

### D.3   Deep Stochastic Binary Networks

The verification of ST estimator in training deep neural networks with mirror descent is conducted on CIFAR-10 dataset[7].

**Model** Our deep SBN model with $L$ binary layers is defined as

$$\boldsymbol{w}^k = \mathrm{sign}(\boldsymbol{\eta}^k - \boldsymbol{\xi}^k), \quad k = 1\ldots L-1 \tag{67a}$$

$$\boldsymbol{x}^k = \mathrm{sign}(\boldsymbol{a}^k(\boldsymbol{w}^k, \boldsymbol{x}^{k-1}) - \boldsymbol{z}^k), \quad k = 1\ldots L, \tag{67b}$$

where $\boldsymbol{a}^k$ are *pre-activations*, *i.e.* linear mappings of preceding layer states $\boldsymbol{x}^{k-1}$ with weights $\boldsymbol{w}^k$. Injected noises $\boldsymbol{\xi}^k$, $\boldsymbol{z}^k$ are independent for all units. The weights $\boldsymbol{w}^k$ in each inner layer are $\pm 1$ Bernoulli with probability $F_\xi(\eta)$. Weights in the first and last layers are real-valued. Pre-activations $\boldsymbol{a}$ consist of a linear operation and batch normalization [28]:

$$\boldsymbol{a}^k = \mathrm{BN}(\mathrm{Linear}(\boldsymbol{x}^{k-1}, \boldsymbol{w}^k)), \tag{68}$$

where Linear is a binary fully connected or convolutional transform and BN has real-valued affine terms (scale, bias) enabled. In several layers also Max Pooling is applied on top. The architecture specification and illustration of the model are given in Fig. 4.

**Initialization** The role of the affine parameters $(\boldsymbol{s}, \boldsymbol{b})$ in BN is to reintroduce the scale and bias degrees of freedom removed by the normalization [28]. In our model these degrees of freedom are important as they control the strength of pre-activation relative to noise. With the sign activation, they could be indeed equivalently represented as learnable bias and variance parameters of the noise since $\mathrm{sign}(x_i s_i + b_i - z_i) = \mathrm{sign}\left(x_i - \frac{z_i - b_i}{s_i}\right)$ assuming $s_i > 0$. Without the BN layer, the result of $\mathrm{Linear}(\boldsymbol{x}^{k-1}, \boldsymbol{w}^k)$ is an integer in a range that depends on the size of $\boldsymbol{x}$. If the noise variance is set to 1, this will lead to vanishing gradients in a large network. With BN and its affine transform, the right proportion can be learned, but it is important to initialize it so that the learning can make progress. We propose the following initialization. We set $s_i = 1$ and $b_i = 0$ (as default for BN)

---

[7] https://www.cs.toronto.edu/~kriz/cifar.html

and *normalize the noise distribution* so that it has zero mean and $F'(0) = \frac{1}{2}$. This choice ensures that the Jacobian $2F'(\boldsymbol{a})$ in Line 5 of Alg. 1 at the mean value of pre-activations is the identity matrix and therefore gradients do not vanish.

We want to initialize weight probabilities $\theta_i = F_\xi(\eta_i)$ as uniform in $[0, 1]$. The corresponding initialization of latent weights is then $\eta_i = F_\xi^{-1}(\theta_i)$ (which would be a completely non-obvious choice to propose empirically for deterministic ST methods).

**Dataset** The dataset consists of 60000 32x32 color images divided in 10 classes, 6000 images per class. There is a predefined training set of 50000 examples and test set of 10000 examples.

**Preprocessing** During training we use standard augmentation for CIFAR-10, namely random horizontal flipping and random cropping of $32 \times 32$ region with a random padding of 0-4 px on each side.

**Optimizer** We use Adam optimizer [31] in all the experiments. The initial learning rate $\gamma = 0.01$ is used for 300 epochs and then we divide it by 10 at epochs 300 and 400 and stop at epoch 500. This is fixed for all models. All other Adam hyper-parameters such as $\beta_1, \beta_2, \varepsilon$ are set to their correspondent default values in the PyTorch [41] framework.

**Training Loss** Let the network softmax prediction on the input image $\boldsymbol{x}^0$ with noise realizations in all layers $\boldsymbol{z}$ be denoted as $p(\boldsymbol{x}|\boldsymbol{z}, \boldsymbol{x}^0)$. The training loss for the stochastic binary network is the expected loss under the noises:

$$\mathbb{E}_{\boldsymbol{x}^0 \sim \text{data}}\big[\mathbb{E}_{\boldsymbol{z}}[-\log p(\boldsymbol{x}|\boldsymbol{z}, \boldsymbol{x}^0)]\big]. \tag{69}$$

The training procedure is identical to how the neural networks with dropout noises are trained [50]: one sample of the noise is generated alongside each random data point.

**Evaluation** At the test time we can either set $\boldsymbol{z} = 0$ to obtain a deterministic binary network (denoted as 'det'). We can also consider the network as a stochastic ensemble and obtain the prediction via the expected predictive distribution

$$\mathbb{E}_{\boldsymbol{z}}[p(\boldsymbol{x}|\boldsymbol{z}, \boldsymbol{x}^0)], \tag{70}$$

approximated by several samples. In the experiments we report performance in this mode using 10 samples. We observed that increasing the number of samples further improves the accuracy only marginally. We compute the mean and standard deviation for the obtained accuracy values by averaging the results over 4 different random learning trials for each experiment.