# OpenReview forum: "Reintroducing Straight-Through Estimators as Principled Methods for Stochastic Binary Networks"
_ICLR.cc/2021/Conference — Reject_

### Official Review · AnonReviewer4 · 2020-10-27
**Unified view and analysis of Straight Through estimators**

**Rating:** 7
**Confidence:** 2

**Review:**


	1. Summary & contributions
This paper analyzes a number of existing straight through (ST) estimators and connects them using a unified view as estimators for stochastic binary networks (SBNs) with Bernoulli weights. Different estimators can be seen as design choices, either relating to the model or optimization procedure. This provides insights in the specific assumptions (implicitly) made by these estimators and allows to analyze their bias and performance.

The paper proposes an extension of ST for deep SBNs to binary weights and conducts experiments to evaluate the quality of ST gradient estimates, showing that the reliability increases with the size of the (boolean) latent space for a VAE, and providing optimization results for a Deep SBN.

	2. Strengths & weaknesses
Strengths:

This paper unifies a number of straight through estimators under a common framework which helps in motivating previously considered 'ad-hoc' rules for gradient backpropagation, and helps understanding of the conditions which may affect the bias/performance of the estimators.

The paper is technical but written in such a way that it is still relatively easy to follow, by keeping notations simple (yet clear) and deferring detailed derivations/proofs to the appendix (which I did not go through in detail).

The paper shows theoretically and experimentally that different design choices for the estimator are possible, as long as different parts (model, initialization and training) are well aligned.

Weaknesses:

As a weakness, the paper feels a bit as an enumeration of related but separate contributions (summed up in the paragraph 'contributions'). The experiments mainly concern section 4 and show somewhat the effect of Bias Analysis IV), but it would improve the paper if (toy) experiments would have been conducted relating to the other analysis in section 2.2 as well, which could then justify the practical relevance of these results.

While the paper motivates the reintroduction of ST as principled methods, the experiments give some insights but do not fully convince of the practical use of the proposed extensions.

It is positive that the paper is largely self-contained, but this makes it a bit difficult to distinguish novel derivations from recapitulations of results from previous work. Also, the paper lacks a discussion section and as a result feels a bit unfinished.

	3. Recommendation
My current recommendation is to accept the paper.

	4. Arguments for recommendation
This paper helps in the theoretical understanding and unifying view of a variety of straight through estimators, which may help advancing these estimators for training networks with binary weights and activations, which is a relevant and difficult problem.

	5. Questions to authors
- Why is a discussion section omitted?
- Figure 1: looking at the y-axis for the top row, it seems that both ARM and ST get worse results with 256 bits than 8 or 64 bits, which hints at underfitting. How does this affect the conclusions?


	6. Additional feedback
Minor comments
- Line 152: where does the symbol f for loss come from?
- Line 305: a reference is missing: ??
- Line 331,5: reminder -> remainder?
- Line 372: ELOB -> ELBO? Or -ELBO (negative ELBO)?
- Some grammar could be improved, e.g. in abstract 'we … obtains, …, explains'

---

> ### Author Response · Authors · 2020-11-18
> **Response to Reviewer4**
>
> Thanks for you detailed comments. We tried to address the pointed weaknesses and questions.
>
> * enumeration of related but separate contributions
>
> We agree that some of the contributions are not in the main focus, but we view them as useful and connected to the common topic of ST estimators.
>
> * it would improve the paper if (toy) experiments would have been conducted relating to the other analysis in section 2.2 as well, which could then justify the practical relevance of these results.
>
> The improved experiments on stochastic auto-encoders now verify more predictions from the analysis 2.2: we compare bias of different variants of ST estimators and demonstrate how it translates to the learning behavior and an accumulated bias. One experimentally validated consequence that is made clear is that identity ST is not suitable for activations but is perfectly suitable for weights, as we derived and explained. We further explained better our conclusions from the deep SBN experiment.
>
> * While the paper motivates the reintroduction of ST as principled methods, the experiments give some insights but do not fully convince of the practical use of the proposed extensions.
>
> At present, we only proposed how to synthesize correct (from the point of view of our derivation) methods that work well for any given noise distribution and latent weights model. We would agree with the reviewer about the Bayesian learning extension, for which we did not demonstrate improved results. We position it rather as assigning a meaning to the combination of latent weights and weight decay. We do not offer experiments with it partially because some toy experiments have been already demonstrated by Meng et al. (2020) and partially because making it work in a larger scale appears to require some further research.
>
> * It is positive that the paper is largely self-contained, but this makes it a bit difficult to distinguish novel derivations from recapitulations of results from previous work.
>
> We try to accurately cite the previous work. One our major motivation (and we believe a contribution as well) is to review this prior work in a clear way and systematize together different aspects.
>
> * (Q1) Discussion section
>
> We extended the discussion of experiments and added the conclusion section
>
> * (Q2) Figure 1: looking at the y-axis for the top row, it seems that both ARM and ST get worse results with 256 bits than 8 or 64 bits, which hints at underfitting. How does this affect the conclusions?
>
> This was indeed the case in the initial submitted version. Somewhat counterintuitive, it was actually very hard to achieve a comparable training loss using more bits (even with a longer training budget). We think the reason for this is the partial posterior collapse in VAE with discrete latent states. We discuss the issue in more detail in Appendix D.1. Basically, increasing the latent space dimensionality did not lead to the solutions that would really make use of these latent bits (more bits are found in collapse) but was making the optimization problem harder. This is the main reason why we switched to plain stochastic auto-encoders, in order to verify the theoretically predicted dependence on the bit length. It is seen now that the experiment in the revision (Fig. 1) does not have an underfitting issue. All y-axis are precisely aligned and with more latent bits lower training loss is reached.
>
> Please let us know if you have further questions / suggestions.

---

> > ### Comment · AnonReviewer4 · 2020-11-21
> > **Thanks for clear answers, no change of rating**
> >
> > Thanks for the clear answers to my questions. I do not consider it necessary to update my rating but I do appreciate the updates to the paper, especially adding a discussion section.

---

### Official Review · AnonReviewer2 · 2020-10-28
**I recommend to accept that paper while my assessment is unconfident due to lacking experties in this area.**

**Rating:** 7
**Confidence:** 2

**Review:**

Summary: The paper presents a principled derivation and analysis of the straight-through (ST) estimator, which is often used to train networks with binary weights and activations.

A strong point of the paper is that the ST estimator is often used to train networks with binary weights and activations. Hence, a more theoretical investigation can be helpful to better understand the ST estimator. Furthermore, the paper seems theoretically sound. However, my assessment is unconfident due to lacking expertise in this area. The empirical experiments confirm that the ST performance improves as the number of latent bits is increased as suggested by the theoretical analysis. Another strong point is that the paper will make code available on Github, which can improve the reproducibility of the experiments.

A weak point of the paper might be its limited potential impact on future works since it mainly provides an analysis of prior empirical ST approaches. Furthermore, a substantial amount of content is only described in the supplementary material. A venue that allows longer submission may be a better fit for this work.

Minor important points:
- Missing reference in Line 305.

---

> ### Author Response · Authors · 2020-11-18
> **Response to Reviewer2**
>
> Thanks for you comments. We would like to say something in response to the mentioned weaknesses.
>
> * Limited potential impact
>
> Please see whether our updated experiments section improves on the potential impact in you view. We try to obtain more experimental insights confirming the analysis and helping to chose and apply ST methods with clarity and awareness of the limitations. Furthermore, we hope our work will facilitate the development of improved methods.
>
> * A substantial amount of content is only described in the supplementary material. A venue that allows longer submission may be a better fit for this work
>
> This is not uncommon for ICLR papers in our experience (e.g. Cong et al., 2019 "GO Gradient for expectation based objectives", which we cite). To be honest, we find it rather hard to publish theoretical results reaching simultaneously a significant contribution, high clarity and a short length. And only the length of the technical part is relaxable in our view. We would appreciate suggestions (from any of the reviewers) for a better publishing strategy / other venues in machine learning that allow longer submissions.
>
> * Missing reference in Line 305
>
> Fixed in the revision. The reference was to the Appendix C.2 "Latent Weight Decay Implements Variational Bayesian Learning"

---

### Official Review · AnonReviewer3 · 2020-10-28
**marginally below acceptance**

**Rating:** 5
**Confidence:** 3

**Review:**

#####################################################

Summary:
This paper presents a systematic way of analysing straight through estimators and derive ST estimators for Stochastic Binary Networks.
#####################################################


Pros:

1. The paper is theoretically strong.
2. The paper covers a large body of relevant work, covering theory behind straight through estimators and stochastic binary networks.



#####################################################


Cons:
1. The novelty of the paper isn't clear, if this paper is an analysis paper, the empirical evidence is weak.
2. The paper is very hard to read, and it is difficult to understand the clear motivation. The paper is too dense while missing any key takeaway point. It misses key details in the experiment section.
3. The utility of the proposed MD estimator is unclear,  it would be helpful if the authors would clarify the interpretation of Table 1 with their write up under "Classification with Deep SBN": the authors state that their method performs as well as the empirical ST, while the table shows it performs worse than their baselines.
4. Overall, the experiment section is scattered, with hard to understand goals. For example, the takeaway from Figure 1 bottom is difficult to understand.


#####################################################


Typos:
Line 305 ??
The paper needs major revisions in terms of notation issues: the vectors should be bold, to distinguish the from scalars.
#####################################################


Questions:
Please address the cons mentioned above.

---

> ### Author Response · Authors · 2020-11-11
> **Clarifications, what we can do**
>
> Dear R3, thanks for you comments.
>
> * Novelty is not clear. Motivation is not clear. Hard to read. Too dense.
>
> We already did our best to present the systematization and the key results in a clear and accessible way. We cannot omit any part of the contribution to make it less dense either in order to devote more space to e.g. motivation.
>
> *  Clarify the interpretation of Table 1. Performs worse than baselines.
>
> Our observations from Table 1 are summarized in lines 456-467. There are two kinds of baselines: ST-based methods and a method of Peters and Welling (2018) also derived for SBNs.
>
> First, we observe from a comparison  in the deterministic mode ('det' column) that our derived ST method is not worse than  empirical ST setups found by trial and error. The differences of 0.1-0.2% can be neglected (it is within an effect of random initialization). The derived ST takes care of the choice of the noise distribution and proper updates and is clearly interpretable. If the same networks we trained are tested in the stochastic mode (10-sample), there is a clear boost of performance, indicating an advantage of SBN models. The takeaway message here is that mirror descent, initialization, and the link between noise and gradient components all work correctly. While some empirical ST methods were already guessed well, they can be replaced now with a more transparent approach.
>
> Second, there is indeed some gap to Peters and Welling (2018) in the stochastic mode (e.g. our 10-sample versus their 16-sample). Nevertheless, the results are very similar considering that the baseline uses a different estimation method, an initialization from a pretrained model and an early stopping (we believe otherwise they would overfit to the relaxation of the SBN considered). The takeaway message here is that ST can be considered in the context of SBN models as a simple but proper baseline. Since we achieve near 100% training accuracy, the optimization fully succeeds and thus the bias of ST is tolerable.
>
> *  The empirical evidence is weak
>
> We plan to update the experiment on auto-encoders (within this review process) to provide a more clear evidence about properties of STE to address concerns and suggestions by R1 and R4. A clear understanding of STE and its properties is the main focus of the paper (both theoretical and experimental).
>
> * Misses key details in the experiment section.
>
> We will appreciate pointing out which details do you find missing relative to the specifications in Appendix D.
>
> *  Notation issues: the vectors should be bold, to distinguish the from scalars
>
> Thanks for bringing this up. We will give a try to the bold notation. At present, we rarely work with scalars unless they are coordinates of vectors such as x_i, in which case the subindex serves as a distinction. We will check consistency of the vector and coordinate-wise notation in the paper. We believe this is a matter of a minor revision in any case.
>
> * Line 305??
>
> Thanks, this refers to the Appendix C.2, where Prop.2 is proven, i.e. this is a part of the present contribution and not an external reference.

---

> > ### Author Response · Authors · 2020-11-24
> > **Remark**
> >
> > Just to make sure, the reviewer finds all relevant answers:
> >
> > We've delivered all the promised improvements discussed above and summarized them in the "Rebuttal Revision" post. In particular we believe we have fully addressed the points "empirical evidence is weak", " Misses key details in the experiment section", "Notation issues".

---

> > ### Comment · AnonReviewer3 · 2020-11-24
> > **Post rebuttal update: No change to score**
> >
> > Thanks to the authors for the response. I appreciate the authors for revising the draft. I am still not convinced of the novelty of this paper from an analysis perspective. The empirical evidence should reflect their theoretical findings, which is still scattered and not connected to the theoretical portion(There are only scattered references to bias analysis IV and V).  Reviewer 4 has pointed out missing simulation studies that could have made the theoretical conclusions more clear. While I lack expertise in this area, an only empirical analysis may not be necessarily indicative of low impact, the conclusions or lessons learnt from such an analysis are the most important deliverables for such a paper, and in my opinion, this paper misses that.  I agree with AR1 that page limits should not be a good excuse to squeeze necessary information into the appendix or not provide a clear gist for the work.

---

> > > ### Author Response · Authors · 2020-11-24
> > > **Still defending**
> > >
> > > We thank the reviewer for an open opinion and giving us an opportunity to still answer. Since all objective points are out of the way, the answer will be rather rhetoric.
> > >
> > > * Scattered references only to bias analysis IV and V
> > >
> > > The analysis part I gives a clearly interpretable properties and understanding on its own, similar to invariances (a part of analysis before I). It is demonstrated on simple examples and does not need experiments to be validated. Analysis parts II and III mainly develop an approach of how at all to say something formally about such crude approximations as ST and are prerequisites for parts IV and V which bring the analysis closer to the questions that are of interest in practice, which are indeed validated in Section 5. And furthermore other practical aspects such as accumulated bias are discussed. We think our overview section 2 (and not the formal details of all the propositions in the appendix) does present the analysis points in a clear and accessible way.
> > >
> > > * Novelty
> > >
> > > When seeing a scarce comment "not convinced of the novelty of analysis" we wonder where is the concrete assessment of all the analysis contributions that we claimed (contributions paragraph). The present review does not reflect anything at all about other parts of the analysis except of Section 2., in particular the explanation of latent weights. These explanations and a formal systematical treatment of ST are the main innovation and contribution of the paper. We can see the practical point of view of the reviewer to some extend, but can the reviewer also think of other readers who would be interested in the analysis in the first place, for the purpose of understanding, using appropriate estimators in appropriate conditions and improving them?
> > >
> > > * most important deliverables
> > >
> > > With all respect we thus disagree that the empirical analysis is the most important deliverable for this paper. There are plenty papers giving empirical analysis of binary networks that cannot define what the ST estimator should be estimating or what the real-valued weights that are later quantized could be doing, mathematically.
> > >
> > > * Clarity of theoretical conclusions
> > >
> > > We have put lots of efforts to make the main paper clear and accessible, including addressing all objective points pointed to us. It takes some efforts from the reader side in turn to digest everything, but it is indeed not as difficult as many papers about biased and unbiased estimators that we cite. Try reading e.g. Cong et al., ICLR 2019 that we cite. Many papers offering empirical studies and improving on benchmarks are much more easy to read, does it mean anything? The related theoretical work of Yin et al. (2019), discussed in Appendix A, analyzes deterministic ST under very unrealistic assumption of training on normally distributed data. We believe our analysis is much more practically relevant, with clear concrete interpretations and experimental outcomes.
> > >
> > > We still do not see what is the reviewer's point about squeezing necessary information into the appendix. Concretely? Appendix contains all necessary formal proofs and details not needed for understanding the work in general but necessary for going into details be it proofs or the details of the experimental setup. How does that prevent accepting a (supposedly) high quality research paper?

---

### Official Review · AnonReviewer1 · 2020-10-31
**The novelty is not high and the writing needs further polishing**

**Rating:** 5
**Confidence:** 3

**Review:**

The paper reintroduces the straight-through estimator with bias-variance analysis. It further discusses its relationship with some constrained optimization methods in convex optimization and

In general, the novelty of the paper on the methodology side is not high. Its value may lie in the theoretical analysis of an existing method. However, the current theoretical analysis is not clear for the following issues:

- On page 3, it says "cannot interchange the gradient and the expectation in z". But z is a continuous variable, why it cannot?
- There is no clear explanation of what does dL(x)/dx mean. x is discrete, so this notation without re-definition is incorrect.
- On page 3, it is unclear what "define now dx/da = 2F'(a)" mean. Why should it be defined in this way?
- On page 4 III), why g(x) is assumed to be Lipschitz continuous?
- On page 8, eq (18), this metric seems strange since the ratio of expectation is not the expectation of ratio. And it only measures the angle between two vectors. Why the L2 norm cannot be used here?

In general, the main paper lacks formal theory statements, while the informal statement does not clearly answer the questions it comes up with.  Another major issue is the writing. The paper does not have a conclusion/discussion part which makes it incomplete. And the main paper has several missing citations/references with (??). In figure 1 caption, it is ELBO not ELOB.  With the concerns listed above, the paper in its current version looks not fully ready for publication.


=====POST-REBUTTAL UPDATES========

Thanks to the authors for the response and the efforts in the updated draft. The updated paper improves writing. The response resolves a part of the queries.  The viewer yet believes the page limits should not be a good excuse to squeeze necessary information into the appendix, otherwise, as AR2 suggests, it may be more proper for other venues.  I raised my rating according to the author's response.

---

> ### Author Response · Authors · 2020-11-10
> **Answers**
>
> Thanks for you questions, they will help us to make the paper more clear and accessible. They are quite simple to resolve. We will answer directly here and incorporate in a cumulative update of the paper (to keep the discussion finite, we want to upload a revision only once).
>
> * On page 3, it says "cannot interchange the gradient and the expectation in z". But z is a continuous variable, why it cannot?
>
> It actually says (line 133) "since it [loss] is not continuously differentiable we cannot...". Indeed, to make an interchange a suitable theorem is needed such as Leibniz integral rule (https://en.wikipedia.org/wiki/Leibniz_integral_rule) that requires the function to be continuously differentiable. The fact that z is a continuous variable is not sufficient. A simple counter-example is that expectation of a step function with a continuous noise density is the noise cdf (and is differentiable) but the gradient of a step function is zero almost everywhere and so its expectation.
>
> * There is no clear explanation of what does $dL(x)/dx$ mean. x is discrete, so this notation without re-definition is incorrect.
>
> In our context $\mathcal{L}$ is defined not by a look-up table for all possible discrete configurations but by composing linear and non-linear functions such as softmax(W2*ReLu(W1 x)). That is, it is defined for any real-valued $x$. We evaluate it only at discrete $x$, but we compute the derivative of the obvious continuous extension at that $x$. Please note (line 154) " provided that $f$ alone is differentiable", here $f$ is a typo and should be $\mathcal{L}$.
>
> * On page 3, it is unclear what "define now dx/da = 2F'(a)" mean. Why should it be defined in this way?
>
> If we define this Jacobian exactly in this way, the estimator that was obtained in (8) can be written as a chain rule (9) and computed using familiar backpropagation.
>
>
> * On page 4 III), why g(x) is assumed to be Lipschitz continuous? For example can L(x) = x^3?
>
> The precise claim in Property III can be used for informal reasoning as follows: if g is Lipschitz continuous and gradients are strong enough (as specified), then the estimator will be good. If it is not Lipschitz continuous, one should consider simplifying the loss function or applying a better estimator.
>
>  * On page 8, eq (18), this metric seems strange since the ratio of expectation is not the expectation of ratio. And it only measures the angle between two vectors. Why the L2 norm cannot be used here?
>
> The equation (18) is motivated in more details in Appendix D.1 (lines 1141-1162). We intend to significantly revise this experiment, including measuring root mean squared error and the expected cosine similarity. The idea was that the expected improvement metric (18) is more natural for use in optimization, but perhaps it is more difficult to present and could be less clear.
>
> * the main paper lacks formal theory statements
>
> Since you write "the main paper", you probably have noticed that all statements made in Section 2 are formally detailed in the appendix B and all formal statements made in Section 3 are proven in Appendix C? The paper is organized in this way in order to give an accessible overview in the main part. We believe the statements in Section 2 are sufficiently precise. Writing all formal statements in the main paper would clutter it and reduce readability. Could you please clarify to us and other reviewers whether the comment is about layout or indeed about a lack of formal statements (which specifically).
>
> * Conclusion/discussion
>
> We will be happy to add a conclusion using the 9th page available now. There was no space for it in 8 pages.
>
> * Novelty is not high
>
> While we do not dare to argue what is a high novelty, should not all the new facts and isights about ST that we obtained by the way of analysis be considered as novelty? There is indeed lots of empirical applications and heuristics around ST (and probably a dozen of new submissions to this ICLR using it), but very little theoretical understanding and no systematic study since the introduction in 2012. Isn't our work timely and strikingly novel in that regard?

---

> ### Author Response · Authors · 2020-11-24
> **Appeal to Reviewer1**
>
> We thank the reviewer for acknowledging receiving our response and raising the rating to "marginally below", as indicated in the updated review.
>
> We nevertheless expect a bit more. If the reviewer agrees with the clarifications given, there remains no objective issues and this can be clearly acknowledged, especially given that many queries were not issues in the first place. If the reviewer disagrees with some specific points, we would appreciate concrete arguments.
>
> We also ask the reviewer to delineate the assessment of the contribution from considerations about paper length / appendices. As we know it, in many topics in ML, papers do have as long technical appendices as they need to, especially ICLR papers. Surely they do not put unnecessary information in there.

---

### Author Response · Authors · 2020-11-17
**Rebuttal Revision**

TL;DR: The revision substantially updates the experiments section. We give more supportive evidence of the theoretical analysis, discuss useful insights and clarify takeaway messages. It also implements all proposed minor improvements of clarity, vector notation, fixing typos and technical inaccuracies. Please look.

----

We thank all the reviewers for their comments and suggestions. We prepared a rebuttal revision of the paper, which we believe addresses well all specific concerns and improves on the mentioned weaknesses. Here is a summary of updates:

* Single hidden layer experiments (autoencoder) updated

We compare now three variants of ST methods (derived ST, identity ST, deterministic ST) with Gumbel-Softmax and ARM. As these are popular variants, it is practically helpful to see their comparison. We evaluate all training losses and measured gradients accuracy using 3 metrics (R1). We are happy to state that the evidence gives multiple confirmations of our analysis, facilitates understanding of biased estimators and has clear practical takeaway messages. The experimental setup is fully detailed in the Appendix B1. One major change that took place is that we switched to plain auto-encoders from variational auto-encoders. Briefly, the (partial) posterior collapse in VAE was compromising our intent to measure dependence on the bit length. We will be happy to discuss this separately. The overall format of the experiment stayed the same though. We believe that for this theoretical work it is an admissible revision. We believe this revision improves on the "weak empirical evidence" (R3), " justify the practical relevance" (R4) "missing any key takeaway point" (R3), "misses key details in the experiment section" (R3), "limited potential impact" (R2), " experiment section is scattered" (R3).

* Training Deep SBNs experiment

We cleaned up Table 1 and reworked the explanations and takeaway points as discussed with R3. Here we try to be more clear and convincing without reworking anything in the experiment. We also detailed more the experimental setup for this part in Appendix B.2 (R3)

* Specific questions discussed with R1 are clarified in the text to improve clarity / accessibility.

* Vectors and matrices are made bold and the conformance of vector and coordinate-wise operations is verified (addressing R3)

* There is a conclusion now

* Fixed typos, technical inaccuracies, proofread

We will follow up with answers to specific questions (e.g. "underfitting" question by R4) that are already resolved in the revision but have not yet been discussed on the board.

Kind regards,
Authors

---

### Decision · Program_Chairs · 2021-01-07
**Final Decision**

**Decision:**

Reject

**Comment:**

This paper collects a variety of results that cast straight-through estimators as arising as principled methods that make a linearization assumption on the loss for functions with binary arguments. R1 & R3 recommended against acceptance, citing clarity concerns and a lack of novelty. R2 & R4 recommended acceptance, but had low confidence. This paper had uncharacteristically low confidence on behalf of the reviewers, and this is my fault. I apologize to the authors for this.

I have read the paper myself. I believe that this paper contains many interesting ideas, but I agree with R1 & R3 that the paper suffers from clarity issues. Unfortunately, these issues persist in the recent revisions, despite having been asked by R1 & R3 to improve the clarity. The authors asked for concrete reference points. Here are some:

- "proxy function" is not well-defined, despite being critical to the arguments.
- deterministic ST is not defined clearly before it is discussed.
-  The section structure of Sec 2 could be improved. At the moment it seems to flow from the loss function to the standard ST algorithm through to a disjointed list of questions addressed in the paper.
- The section titles are not particularly informative.
- It is difficult to know which results are known and which results are new.

In general, I believe this work could benefit from a significant restructuring. It would be best to delineate preceding work in its own section, then lay out the new results, making sure that all of the important concepts are clearly defined. I think many of these results are valuable for the community, but the current draft makes it challenging for these great ideas to reach their full potential.